# Toll signalling controls intestinal regeneration in *Drosophila*

Aiswarya Udayakumar[1], Filippos Stavropoulos[2], Theodosia Hadjipanteli[2], Guofan Peng[3], Shivohum Bahuguna[4], Caitlin MacClay[1], Jeffrey Y. Lee[1], Qi Xiao[5], Yuxian Xia[5], Michael Boutros[4], Jun Zhou[3], Yiorgos Apidianakis[2], Chrysoula Pitsouli[2,*] and Petros Ligoxygakis[1,*]

## ABSTRACT

The intestinal interphase is where epithelial renewal and tissue maintenance are balanced alongside immunological regulation. How these functions integrate with cellular signalling is under investigation. Here, we studied the role of the evolutionarily conserved innate immune Toll/NF-κB pathway in *Drosophila* intestinal regeneration. We found that the core components of the canonical Toll pathway were necessary for intestinal stem cell (ISC) mitosis in homeostasis and upon infection. Toll activation was sufficient to push ISCs into mitosis and the enteroblast (EB) fate, but blocked EB differentiation resulting in ISC and EB accumulation. This was mediated by JNK and Akt/TOR signalling. When JNKK, JNK, Akt or TOR activity was reduced in gut progenitors, ISC mitosis was suppressed. Toll activation also triggered suppression of antimicrobial lysozyme and amidase genes, which led to increased gut bacterial density. Our results identify Toll as necessary and sufficient for ISC mitosis. Our model is that the Toll pathway acts as a regulator of the intestinal landscape integrating JNK and Akt signals to achieve gut tissue renewal and control of commensal bacteria density.

KEY WORDS: Toll, *Drosophila*, Gut, Intestinal stem cells

## INTRODUCTION

Toll-like receptors (TLRs) are pattern recognition receptors (PRRs) that recognize conserved microbial molecules like lipopolysaccharide (LPS), bacterial DNA, bacterial peptidoglycans, zymosan and beta-glucan from fungi, and single-stranded and double-stranded RNA from viruses (Takeda and Akira, 2015). The function of Toll and TLRs in immunity was first described in *Drosophila* (Lemaitre et al., 1996). In flies, Toll drives systemic immune responses [including secretion of antimicrobial peptides (AMPs)] following fungal infection. This discovery, along with the identification of TLR4 as the LPS receptor in mice (Poltorak et al., 1998), became central in the notion of the evolutionary conservation of Toll-mediated innate immune responses in vertebrates and invertebrates. The *Drosophila* Toll pathway is activated by the proteolytically cleaved form of a cytokine-like ligand, the Nerve Growth Factor homologue Spaetzle (Spz) (Hepburn et al., 2014). This means that, unlike its mammalian counterparts, *Drosophila* Toll does not interact directly with microbial molecules (Gangloff et al., 2003). As such, pathogen recognition in flies occurs upstream of Toll, with the participation of the Peptidoglycan Recognition Protein SA (PGRP-SA) (Michel et al., 2001).

On Spz binding, the activated dimeric Toll receptor binds via its TIR domain to the adaptor protein MyD88. MyD88 recruits another adaptor protein, Tube (orthologue of human IRAK4) and the Pelle kinase (orthologue of human IRAK1) to form a MyD88-Tube-Pelle complex via interaction with their Death Domains (DD). MyD88 and Pelle bind to each other through separate DDs on Tube (Sun et al., 2002, 2004). The complex is then responsible for phosphorylation and eventual degradation of the IκB factor, Cactus (Geisler et al., 1992). Under non-signalling conditions, Cactus is bound to either of the NF-κB factors, Dorsal and/or Dif (Steward, 1987; Manfruelli et al., 1999; Rutschmann et al., 2000). After dissociation from Cactus, Dorsal/Dif is free to undergo nuclear translocation and mediate the expression of a set of target genes to establish dorsal-ventral polarity in the embryo (NF-κB/Dorsal) or regulate host defence during infection (NF-κB/Dif) (Steward, 1987; Manfruelli et al., 1999; Rutschmann et al., 2000).

However, any potential role of Toll/TLRs in the regeneration of the intestinal epithelium is still under investigation. Results from mammalian models are somewhat contrasting. For example, one of the first studies in the field by Rakoff-Nahoum et al., concluded that MyD88-deficient mice had increased intestinal epithelial cell (IEC) proliferation due to TLR signalling disruption. Similar results were later observed in TLR1- and TLR9-deficient mice (Rakoff-Nahoum et al., 2004; Rose et al., 2012). Further, activating TLR4 using LPS supressed ISC proliferation and induced cell death in organoids (Neal et al., 2012; Naito et al., 2017). Conversely, Tlr4⁻/⁻ organoids were protected from apoptosis induced by LPS exposure. Similar findings were also observed in *in vivo* studies conducted on mono-colonized gnotobiotic mice (Neal et al., 2012; Naito et al., 2017). These results show that on activating TLR signalling, there is reduced cell proliferation and higher apoptosis, thus inhibiting epithelial cycling and renewal. Contradictory evidence indicates increased IEC proliferation in small intestines and colon on TLR4 activation (Sodhi et al., 2012; Santaolalla et al., 2013). Further, TLR4 deficiency has been shown to downregulate proliferating cell nuclear antigen (PCNA), a marker that reflects cell turnover rates in IECs of the small intestine (Sodhi et al., 2012; Santaolalla et al., 2013).

### The *Drosophila* midgut

In this complex background, studying Toll signalling in the *Drosophila* gut seems perhaps an obvious choice. Nevertheless,

[1]Department of Biochemistry, University of Oxford, South Parks Rd, Oxford, OX1 3QU, UK. [2]Department of Biological Sciences, University of Cyprus, 1 University Avenue, 2109 Aglantzia, Nicosia, Cyprus. [3]Hunan Key Laboratory of Animal Models and Molecular Medicine, The Affiliated XiangTan Central Hospital of Hunan University, School of Biomedical Sciences, Hunan University, Changsha 410082, Hunan Province, People's Republic of China. [4]German Cancer Research Centre (DKFZ), Division Signalling and Functional Genomics, Heidelberg University, D-69120 Heidelberg, Germany. [5]Key Laboratory of Gene Function and Regulation Technology, Genetic Engineering Research Centre, School of Life Sciences, Chongqing University, Chongqing 401331, People's Republic of China.

*Authors for correspondence (pitsouli.chrysoula@ucy.ac.cy; petros.ligoxygakis@bioch.ox.ac.uk)

P.L., 0000-0002-9498-9993

we know little about the role of Toll in the renewal of the adult intestinal epithelium. This is even more surprising considering the extensive knowledge on the regulation of the fly intestinal epithelium in both homeostatic as well as stress conditions. In *Drosophila*, intestinal stem cells (ISCs) give rise to all other cell-types of the adult gut. The posterior midgut is maintained by ~1000 ISCs that are scattered amongst 10,000 posterior midgut cells (reviewed by O'Brien and Bilder, 2013; O'Brien, 2022). They express the Notch ligand Delta (Dl) which is generally used to identify the ISC population (Ohlstein and Spradling, 2006). ISCs divide asymmetrically to self-renew and produce transient enteroblasts (EBs) or enteroendocrine progenitors (EEPs), which give rise to enterocytes (ECs) or enteroendocrine cells (EEs), respectively. The differentiating ISCs that express *prospero* (*pros*) form the EEPs and EEs, while the ones that do not express *pros* form the ECs via the transient EB state, with high levels of Notch signalling (Micchelli and Perrimon, 2006; Ohlstein and Spradling, 2007; Guo and Ohlstein, 2015). EBs are diploid and express the transcriptional regulator of the Notch pathway, Suppressor of Hairless [Su(H)]. The ISCs, EBs and EEPs form the progenitor cell population, while the ECs and EEs form the terminally differentiated cell types. Progenitor cells express the transcription factor Escargot (Esg), which is used for progenitor cell identification and labelling (Micchelli and Perrimon, 2006). In a manner analogous to the mammalian gut, the subregions of the *Drosophila* gut show variation in ISC cycling rates and expression of markers like Dl (Strand and Micchelli, 2011), justifying the differences in regional physiology with variation in stem cell characteristics. Studies also show that these subregions are also re-established following tissue injury (reviewed by O'Brien and Bilder, 2013).

ISCs exhibit increased turnover and participate actively in tissue regeneration (Micchelli and Perrimon, 2006; Liang et al., 2017). ECs constitute the absorptive cell population and make up 90% of the differentiated gut cells (Biteau and Jasper, 2014). They are involved in metabolism by secretion of digestive compounds, nutrient absorption, transport and immune responses. They are morphologically distinct from other cell populations due to their large polyploid nuclei. ECs express the gene *Myosin31DF* [or *Myosin 1A* (*Myo1A*)], used for identification. EEs comprise less than 10% of cell population of the gut (Biteau and Jasper, 2014; Zeng and Hou, 2015; Chen et al., 2016). They secrete hormones and peptides that control functions like gut motility and inter-organ communication involved in the gut-brain axis to modulate metabolism and behaviour in response to nutrient availability while also providing antimicrobial defence (Guo et al., 2019; reviewed by Guo et al., 2022). Pros is the genetic marker for EEs and *pros* depletion provokes EE de-differentiation back to ISCs (Guo et al., 2024). ECs and EEs thereby orchestrate antimicrobial responses as performed by PCs and goblet cells in mammals (Chen et al., 2016; reviewed by Drucker, 2016). Beyond ECs (Bahuguna et al., 2022), single-cell RNA-sequencing studies have found low expression of Toll in ISCs (Hung et al., 2020) and moderate expression in EEs (Hung et al., 2020; Shin et al., 2022).

We have examined the possible implication of *Drosophila* Toll in intestinal regeneration. We have found that the Toll signalling axis is needed in ISCs for epithelial renewal both in homeostatic conditions and upon infection. Toll activation in adult gut progenitor cells led to ISC mitosis and blockage of EB differentiation. This resulted in intestinal dysplasia with a dramatic increase in total progenitor cell numbers through JNK signalling and Akt/TOR activity. Moreover, Toll activation led to an increase in gut bacterial density through the transcriptional suppression of antimicrobial lysozymes and amidases. We propose that Toll functions as a 'leash' to regulate the intestinal landscape in health and disease.

## RESULTS

To avoid developmental effects and assess the role of the Toll pathway only in adult intestinal cells, we used the UAS/GAL4 system (Brand and Perrimon, 1993) combined with the temporal control offered by the use of GAL80ts (Suster et al., 2004). We used $ISC^{ts}$-GAL4 [esg-GAL4,Su(H)-GAL80,tub-GAL80ts,UAS-GFP] (Zhai et al., 2015), $EB^{ts}$-GAL4 [Su(H)-GAL4, tub-GAL80ts, UAS-Histone-RFP or UAS-CD8GFP] (Zeng et al., 2010), $EC^{ts}$-GAL4 (Myo1A-GAL4,tub-GAL80ts, UAS-GFP) (Jiang et al., 2009), $EE^{ts}$-GAL4 (prosV1-GAL4, UAS-GFP) (Siudeja et al., 2021) and $esg^{ts}$-GAL4 (esg-GAL4, tub-GAL80ts, UAS-GFP) (Micchelli and Perrimon, 2006), to drive expression of GAL4 in ISCs, EBs, ECs, EEs and all progenitors (ISCs, EBs and EEPs), respectively. To quantify cells positive for various signals [green fluorescent protein (GFP), fluorescently labelled antibodies] we generated a machine learning sequence based on a generalist cell segmentation algorithm (Stringer et al., 2021, see Materials and Methods).

### Toll is required in ISCs for intestinal mitosis in homeostasis and upon pathogenic infection

To explore the potential role of the Toll pathway (simplified schematic in Fig. 1A) in the adult intestine, we expressed RNAi lines targeting genes encoding the core components of the pathway like the receptor Toll, and the downstream transcription factors Dorsal and Dif in intestinal progenitors (ISCs/EBs/EEPs) and in mature ECs and EEs using $esg^{ts}$-GAL4, $EC^{ts}$-GAL4 and $EE$-GAL4, respectively. Then, we measured the midgut mitotic index of sucrose-fed uninfected flies and flies orally infected with the Gram-negative bacterium *Pseudomonas aeruginosa* (strain PA14). Oral *P. aeruginosa* infection is lethal and induces a significant increase in intestinal mitosis compared to sucrose-fed controls (Apidianakis and Rahme, 2009). We stained midguts with the anti-phospho-Histone H3 (pH3) antibody, quantified the mitotic pH3+ intestinal cells and confirmed that *P. aeruginosa* infection increased progenitor cell mitosis by ~10-fold compared to sucrose feeding, compared to the control genotype tested (Fig. 1B). However, when components of the Toll pathway, including *Toll* (also known as *Toll-1*), *dorsal* and *Dif* were depleted by RNAi from intestinal progenitors of 10-day-old flies, mitosis following sucrose treatment as well as infection was significantly reduced (Fig. 1B). In contrast, silencing Toll pathway components in mature ECs (Fig. 1C) or EEs (Fig. 1D) had no significant impact on intestinal mitosis (with the exception of dorsal silencing in EEs that exhibited somewhat reduced mitosis upon infection). Therefore, under homeostatic and regenerative conditions, the Toll pathway is necessary for mitosis in intestinal progenitors, but not mature ECs and EEs.

To assess whether Toll is necessary in ISCs or EBs (both of which express *esg* and contribute to the gut progenitor population), we quantified mitosis after silencing the core components of the pathway namely, *Toll*, *dorsal* and *Dif* specifically in adult ISCs and EBs using the $ISC^{ts}$-GAL4 and $EB^{ts}$-GAL4, respectively (Fig. 1E,F). We found that silencing Toll signalling components in ISCs (Fig. 1E) phenocopies the reduction in mitosis observed with $esg^{ts}$-GAL4 in both homeostatic and regenerative conditions. Furthermore, Toll pathway component silencing in EBs significantly reduces mitosis in infected but not uninfected midguts (Fig. 1F). Overall, our data show that the Toll pathway is absolutely required for mitosis specifically in ISCs under homeostatic and regenerative conditions, while having a role in EB mitosis only upon infection.

To further address the role of Toll pathway in ISCs during homeostatic conditions, we silenced additional components of the pathway in ISCs without any treatments, and we measured the

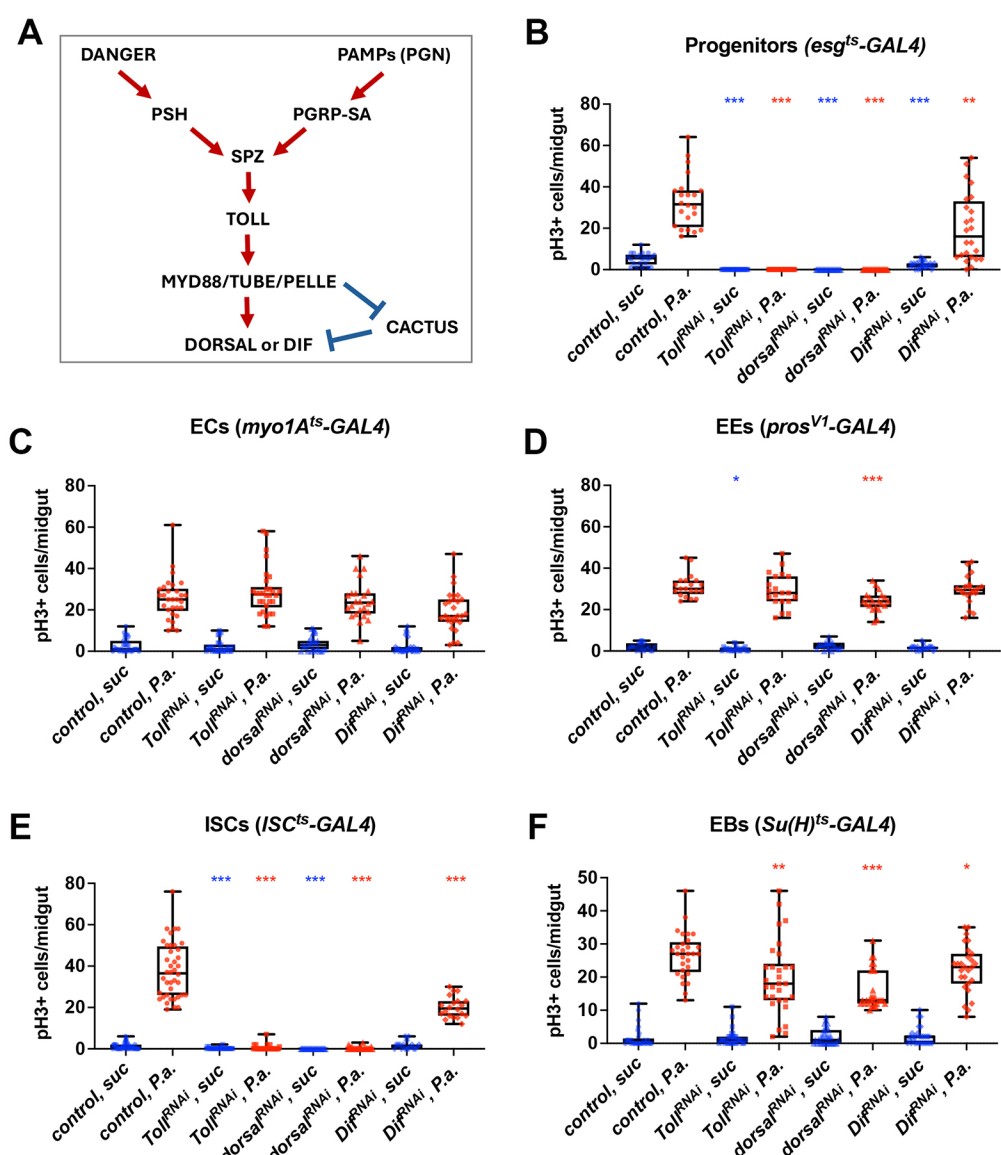

**Fig. 1. The Toll pathway is required in intestinal progenitors for midgut mitosis following infection.** (A) Toll pathway simplified: danger (e.g. pathogen proteases) or pathogen associated molecular patterns [PAMPs, e.g. peptidoglycan (PGN)] activate the PSH protease or the Peptidoglycan Receptor Protein-SA (PGRP-SA) and, through a protease cascade, cleave and activate SPZ, the Toll ligand. A receptor adaptor complex (Myd88/Tube/Pelle) targets the IκB homologue (Cactus) for degradation and thus enables the NF-κB homologue Dif or Dorsal to enter the nucleus and regulate target genes. (B-F) Quantification of the mitotic index of adult midguts upon tissue-specific silencing of *Toll*, *dorsal*, *Dif* and *spz* in uninfected (suc, blue) conditions and upon pathogenic infection with *P. aeruginosa* PA14 (P.a., red) in progenitors using *esg^ts-Gal4* (B), in ECs using *Myo1a^ts-Gal4* (C), in EEs using *pros^V1-Gal4* (D), in ISCs using *ISC^ts-Gal4* (E) and in EBs using *Su(H)^ts-Gal4* (F). The graphs correspond to sums of three experiments (n≥30). Unpaired Student's *t*-test was used for pairwise comparisons between control and silenced genotypes in uninfected and *P.a.*-infected conditions (\*$P<0.05$, \*\*$P<0.01$, \*\*\*$P<0.001$). Box plots show values of individual flies (dots) with the middle line being the mean and the box indicating the first and third interquartile range. Whiskers show the whole range of values.

mitotic index as well as the ISC numbers in the gut of 10-day-old flies. *ISC^ts-GAL4*-driven RNAi targeting Toll, its ligand Spz, the upstream receptor PGRP-SA and the NF-κB homologue and downstream transcription factor Dif, led to significant reduction in mitosis as well as ISC numbers (GFP⁺ cells) compared to controls (Fig. S1A tissue images and Fig. S1B,C quantification). Conversely, to assess whether the Toll pathway is sufficient for mitosis, we overexpressed *UAS-Toll^10b* and *UAS-Dif* specifically in ISCs. *Toll^10b* is a constitutively active form of the Toll receptor that encompasses an amino acid replacement (C781Y) on the external surface of the extracellular part of the protein near the start of the transmembrane domain (Anderson et al., 1985; Schneider et al., 1991). We found that constitutive activation of the pathway in ISCs increased the number of mitotic (pH3⁺) cells and ISCs (GFP⁺) (Fig. S1A tissue images and Fig. S1B,C quantification for pH3⁺ and GFP⁺, respectively).

The requirement for Toll and NF-κB/Dif in intestinal mitosis was also indicated when using a stress agent, namely the detergent dextran sulphate sodium (DSS). DSS damages the intestinal epithelium and initiates ISC proliferation and regeneration (Jiang et al., 2011). As expected, DSS treatment induced intestinal mitosis in the *esg^ts-GAL4* genetic background (Fig. S1D). However, depletion by RNAi of Toll

or NF-κB/Dif in intestinal progenitors (*esg^ts-GAL4*) resulted in a significant reduction of pH3⁺ cells compared to the control DSS treatment (Fig. S1D). Moreover, expression of Toll and Spz also localized in progenitor cells, as visualized by using a GFP insertion (MiMIC line; see Materials and Methods) into their native locus (Fig. S1E). This was also the case with GFP insertion of *cactus*, *Dif* and *Dorsal* (Fig. S1E). Taken together, the above experiments show that, following infection, stress damage as well as homeostatic conditions, the classical Toll pathway is necessary and sufficient to control intestinal mitosis and ISC numbers in adult *Drosophila*.

Since we found Toll important for maintenance of adult ISC numbers as indicated above, we tested whether loss of Toll in intestinal progenitors had an impact on host survival. As expected, silencing Toll in the fat body of female flies (*yolk-GAL4*) made them significantly more susceptible to *Staphylococcus aureus* systemic infection (injection) compared to controls ($P<0.0001$, LT₅₀ 36 h for Toll-RNAi versus 48 h for controls; Fig. S2A). *S. aureus* oral infection increased the survival time scale of the host from hours to days, showing that the gut was an important barrier. However, there was still a significant difference between Toll silenced in progenitor cell versus controls, indicating the significance of Toll in

intestinal progenitors following *S. aureus* infection (*P*<0.0001, LT$_{50}$ 15.7 days for Toll-RNAi versus 23.3 days for controls; Fig. S2B). Indeed, *S. aureus* triggered Toll-dependent progenitor expansion (Fig. S2C,D). Toll overactivity in intestinal progenitor cells (no infection) also significantly curtailed lifespan (*P*<0.0001, LT$_{50}$ 37 days for Toll-RNAi versus 45 days for controls; Fig. S2B). These results indicate that Toll is important for host survival following infection in progenitor cells but its constitutive activity without infection also reduced host survival.

## Multiple tissue sources of Spz influence mitosis at the intestinal interphase

Spz, the ligand of Toll, is a secreted molecule and therefore it is possible that its source tissue may not be limited to the ISCs or the gut. To assess the source of the Toll ligand, we silenced *spz* specifically in

intestinal cells, such as all progenitor cells, ISCs, EBs, ECs, EEs as well as in the trachea, which oxygenates and communicates with the gut to control intestinal mitosis (Tamamouna et al., 2021; Perochon et al., 2021) and the fat body, which is a key tissue involved in systemic anti-microbial response and secretory functions (Yu et al., 2022). We used two independent and effective RNAi lines targeting *spz* (see Materials and Methods) to silence the gene in different cell populations (Fig. S3). As expected, infection with *P. aeruginosa* activated intestinal mitosis in control flies (Fig. 2A-G). However, when *spz* was silenced, intestinal mitosis was reduced. Specifically, we found that, following infection, both lines were able to significantly reduce the number of pH3$^+$ cells in progenitor cells (Fig. 2A), ISCs (Fig. 2B), EBs (Fig. 2C) and ECs (Fig. 2D), but not in EEs (Fig. 2E). Beyond the gut, silencing *spz* in the trachea (Fig. 2F) and the fat body (Fig. 2G), both *spz*-RNAi lines significantly reduced

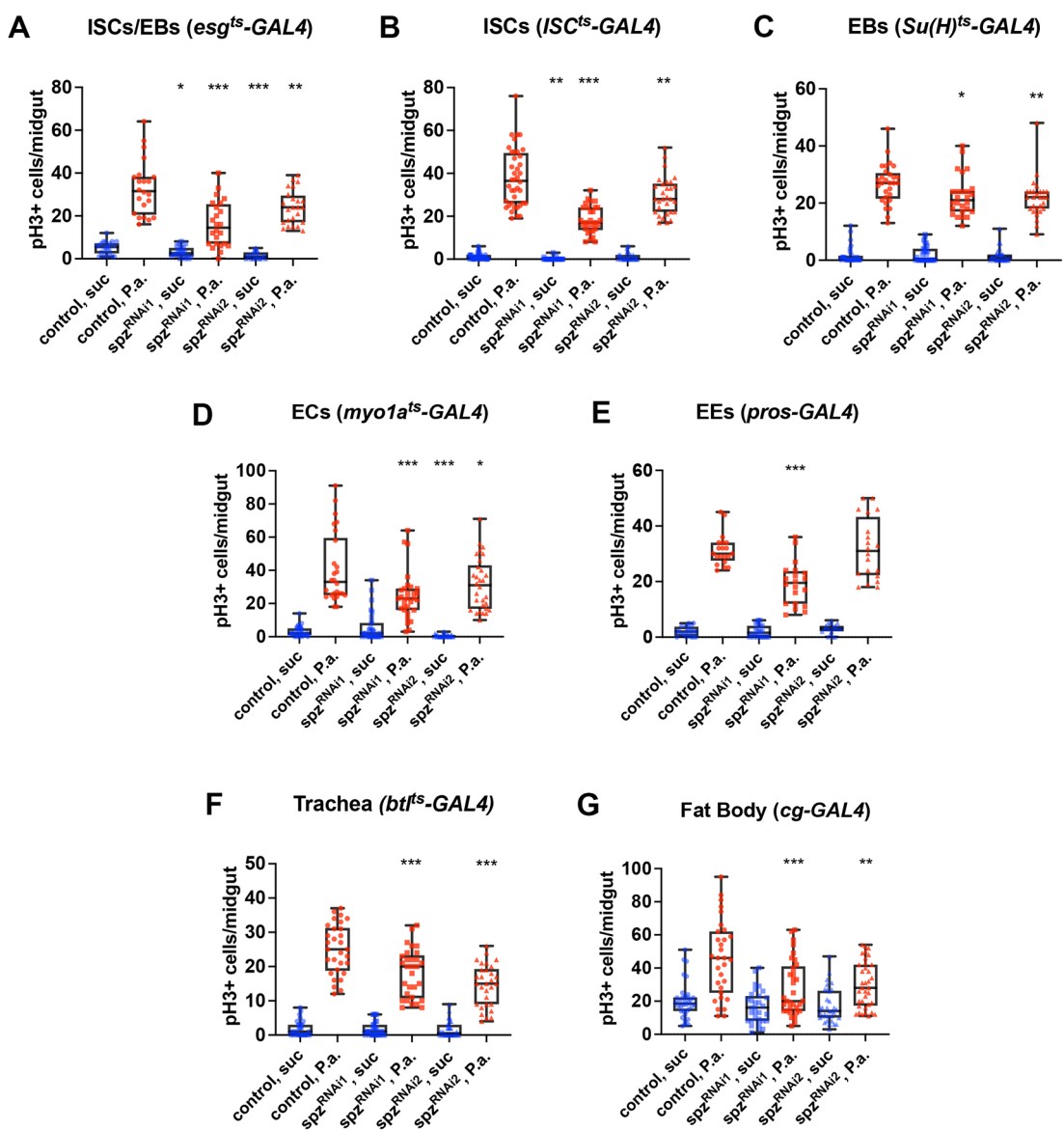

**Fig. 2. The Toll ligand Spz is required in various midgut cell types and systemically for midgut mitosis.** (A-G) Quantification of the mitotic index of adult midguts in uninfected (suc, blue) and in *P. aeruginosa*-infected conditions (P.a., red) upon tissue-specific *spz* silencing in the intestinal progenitors (*esg$^{ts}$-Gal4*; A), the ISCs (*ISC$^{ts}$-Gal4*; B), the EBs [*Su(H)$^{ts}$-Gal4*; C], the ECs (*myo1A$^{ts}$-Gal4*; D), the EEs (*pros$^{V1}$-Gal4*, E); the trachea (*btl$^{ts}$-Gal4*; F) and the fat body (*cg-Gal4*; G) using two independent RNAi lines (spz$^{RNAi1}$ and spz$^{RNAi2}$). The graphs correspond to sums of at least two experiments (*n*≥20). Unpaired Student's *t*-test was used for comparisons between control and silenced genotypes in uninfected and *P.a.*-infected conditions (**P*<0.05, ***P*<0.01, ****P*<0.001). Box plots show values of individual flies (dots) with the middle line being the mean and the box indicating the first and third interquartile range. Whiskers show the whole range of values.

the amount of pH3+ cells in the gut. In contrast, when flies were given sucrose, silencing *spz* in progenitors (Fig. 2A, both lines), ISCs (Fig. 2B, one line) and ECs (Fig. 2D, one line) was able to reduce pH3+ cell numbers, but not in the trachea (Fig. 2F) or the fat body (Fig. 2G). Taken together, these results suggest that, following infection, multiple sources of Spz contribute to regulate mitosis in the intestinal interphase but under homeostatic conditions (sucrose) only cells in the gut are important for sourcing the Toll ligand.

## Constitutive Toll activation increases ISC mitosis while blocking EB differentiation

Our previous results established that *Toll10b* overexpression in ISCs was sufficient to induce mitosis in homeostatic conditions, a phenotype reminiscent of regeneration upon pathogenic infection. Using *esg^ts-GAL4*, we expressed *Toll10b* in progenitor cells and quantified the midgut mitotic index as well as the numbers of different intestinal cells in *esg^ts-GAL4 UAS-Toll10b* compared to *esg^ts-GAL4* crossed to the *w1118* genetic background as controls. Progenitor cells were labelled with green fluorescence through a *UAS-GFP* transgene. On Toll activation, we observed an expansion in the number of GFP+ cells in 10-day-old guts (Fig. 3A). This was accompanied by a significant increase in pH3+ cells (Fig. 3B).

Using quantitative image analysis (see Materials and Methods), we measured the proportions of different cells according to number, size and presence or absence of GFP comparing *Toll10b* guts to controls. We observed that: (1) There was a massive expansion in the total cell population, resulting in significantly higher cell number per frame compared to control (Fig. 3C); (2) The proportion of GFP+ cells increased, with more than 90% of the cells in the R4 region being GFP+ (Fig. 3D); (3) The proportion of small nuclei cells increased (Fig. 3E); (4) The proportion of large nuclei cells decreased (Fig. 3F).

These measurements indicate that the massive increase in the total number of cells in *Toll10b* flies may be explained by both an increase in the rate of ISC proliferation and/or by blocked EB differentiation (and thus accumulation of GFP+ cells). The latter conclusion was supported further by the decrease in the proportion of large nuclei cells (ECs) and the increase in the proportion of cells with small nuclei (ISCs/EBs/EEs).

To assess whether the observed mitosis is restricted to ISCs, we used the ISC-only *Dl-lacZ* reporter (Ohlstein and Spradling, 2007). We expressed *Toll10b* via *esg^ts-GAL4; Dl-lacZ* and followed ISCs with anti-β-galactosidase (*lacZ*) staining. Constitutive activation of Toll increased the number of *Dl-lacZ*+ cells indicating ISC

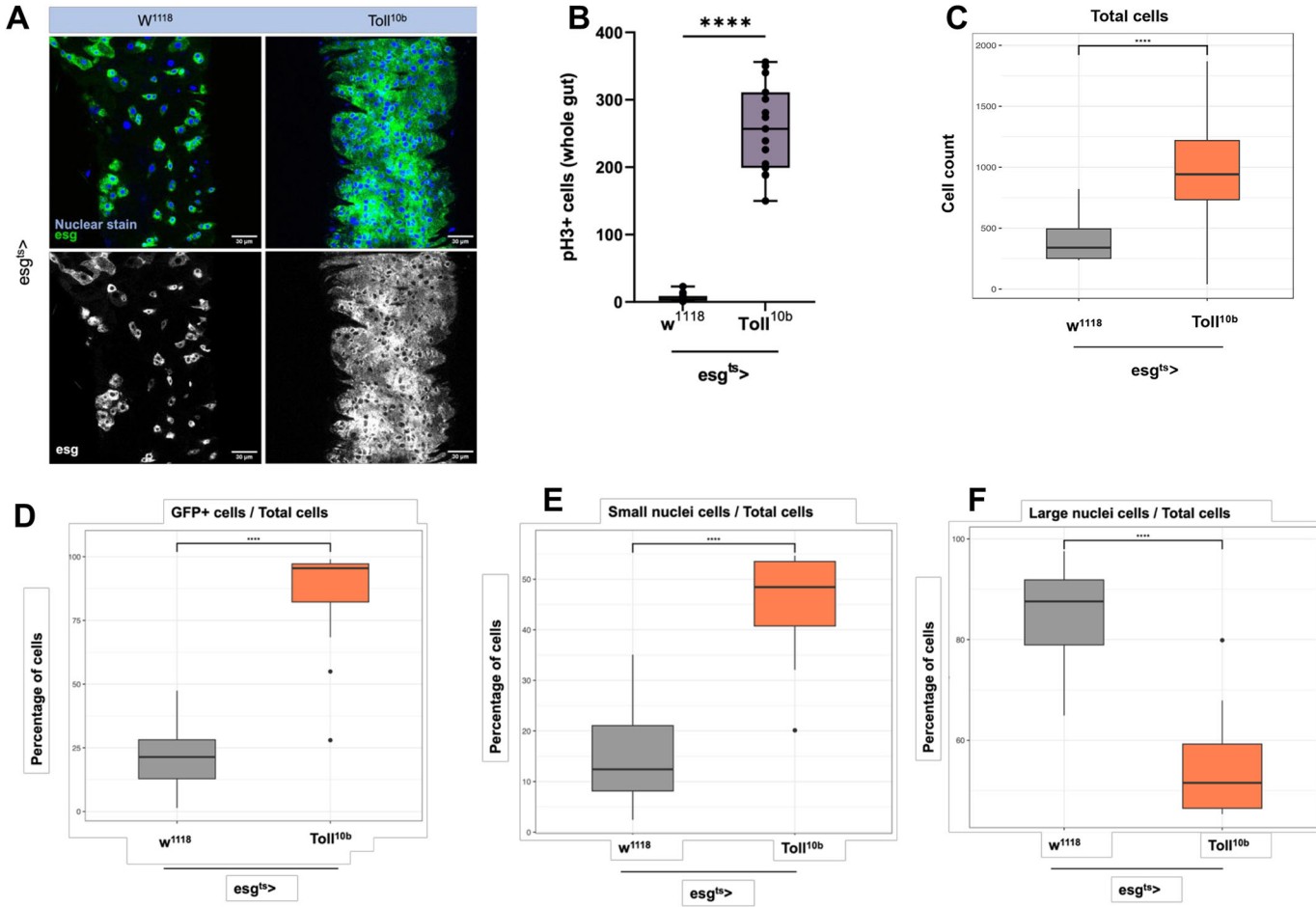

**Fig. 3. Activation of Toll signalling in progenitor cells increases intestinal mitosis.** (A) A single frame from 2 µm thick *z*-stacks obtained from the R4 region of 10-day-old midguts using confocal imaging. ISCs and EBs are green (GFP+), and nuclei are blue. Scale bars: 30 µm. (B) pH3+ counts from whole guts of 10-day-old flies. (C) Total number of cells. (D) Proportion of GFP+ cells. (E) Proportion of small nuclei. (F) Proportion of large nuclei. Genotypes in all panels are control (*esg^ts* crossed with *w1118* and *esg^ts* crossed with *Toll10b*). Percentages were statistically compared using unpaired Student's *t*-test (****P<0.0001). Box plots show values of individual flies (dots) with the middle line being the mean and the box indicating the first and third interquartile range. Whiskers show the whole range of values. Images shown in A are the same as in Fig. 5A, as experiments in Figs 3 and 5 were carried out in parallel using the same controls.

expansion (Fig. 4A). Co-localization of Dl expression and the pH3+ signal would indicate that mitosis induction under active Toll signalling only involved ISCs and instead of intestinal mitosis we would be able to specify the increase in pH3+ cells as ISC mitosis. Indeed, all pH3+ cells were also *Dl-lacZ+* therefore identifying ISC mitosis as one reason for the significant increase of progenitor (GFP+) cells (Fig. 3A). Moreover, expressing *Toll10b* only in ISCs, using the *Dlts-GAL4*, resulted in significantly increased pH3+ counts (Fig. 4B). In contrast, the number of pH3+ cells when *Toll10b* was expressed only in EBs was statistically indistinguishable from controls (Fig. 4C), while EBs [marked by activation of *Su(H)*] increased (Fig. 4D). This meant that activation of Toll triggered mitosis in ISCs but not in EBs. Nevertheless, the fact that EB-only expression of *Toll10b* increased EB numbers without mitosis (Fig. 4D), indicated accumulation of EBs because their differentiation to ECs was blocked.

Compared to control, *Toll10b* expression in progenitor cells slightly increased the absolute number of EEs as measured by quantifying the Pros+ cells (Fig. S4A). However, the ratio of Pros+/total cells was decreased (Fig. S4B), while the ratio of GFP+ and Pros+/Pros+ cells was increased (Fig. S4C). This indicated either mis-differentiation of EEs or blocking EEPs from differentiating to EEs (hence the accumulation of cells, both GFP+ and Pros+, compared to just Pros+ cells).

Taken together, the data confirm that the massive expansion of GFP+ cells in the intestinal epithelium following sustained Toll signalling in progenitors involves two major phenomena: (1) induction of ISC mitosis and (2) the blockage of EBs in their path to ECs. Both of these processes result in the accumulation of GFP+ cells. In addition, some of the GFP+ cells represent blocked EEPs or mis-differentiated EEs.

## The canonical Toll pathway mediates effects of Toll[10b] activity in progenitor cells

To ascertain whether signalling components of the canonical Toll pathway mediated the effects described above we activated Toll (*Toll10b*) in conditions of *Dif* depletion via RNAi. This resulted in a significant reduction of progenitor (Fig. 5A) and pH3+ cells (Fig. 5B). This reflected suppression of total cell numbers (Fig. 5C) as well as the proportion of GFP+ cells in relation to total cells (Fig. 5D). Conversely, *cactus* RNAi in progenitor cells significantly increased the numbers of pH3+ cells (Fig. 5E), while expression of an active form of the ligand Spz also triggered a pH3+ increase (Fig. 5F). These results show that the canonical Toll pathway is involved in regulating proliferation of the intestinal epithelium, with Spz acting as a mitogen.

## Global transcriptional changes in guts with constitutive Toll activity

We next explored the global intestinal transcriptional network in which Toll signalling integrates in progenitor cell transcription. We isolated 20-day-old guts and performed bulk transcriptomics (RNA-seq) to explore the gene expression landscape in 20-day-old guts of *esgts-GAL4* (crossed to *w1118*), *esgts-GAL4/UAS-Toll10b* and *esgts-GAL4/UAS-Toll10b; UAS-bskDN* flies. BskDN is a dominant negative form of JNK, a pathway implicated in ISC mitosis (Biteau et al., 2008). Principal component analysis identified the transcriptomes of *esgts-GAL4* control and *esgts-GAL4/UAS-Toll10b* guts as significantly distinct, while *esgts-GAL4* controls and *esgts-GAL4/UAS-Toll10B; UAS-bskDN* displayed some shared characteristics (Fig. S5A). This was also evident in comparisons of differentially expressed genes (DEGs) >log2< when comparing *esgts-GAL4* controls and

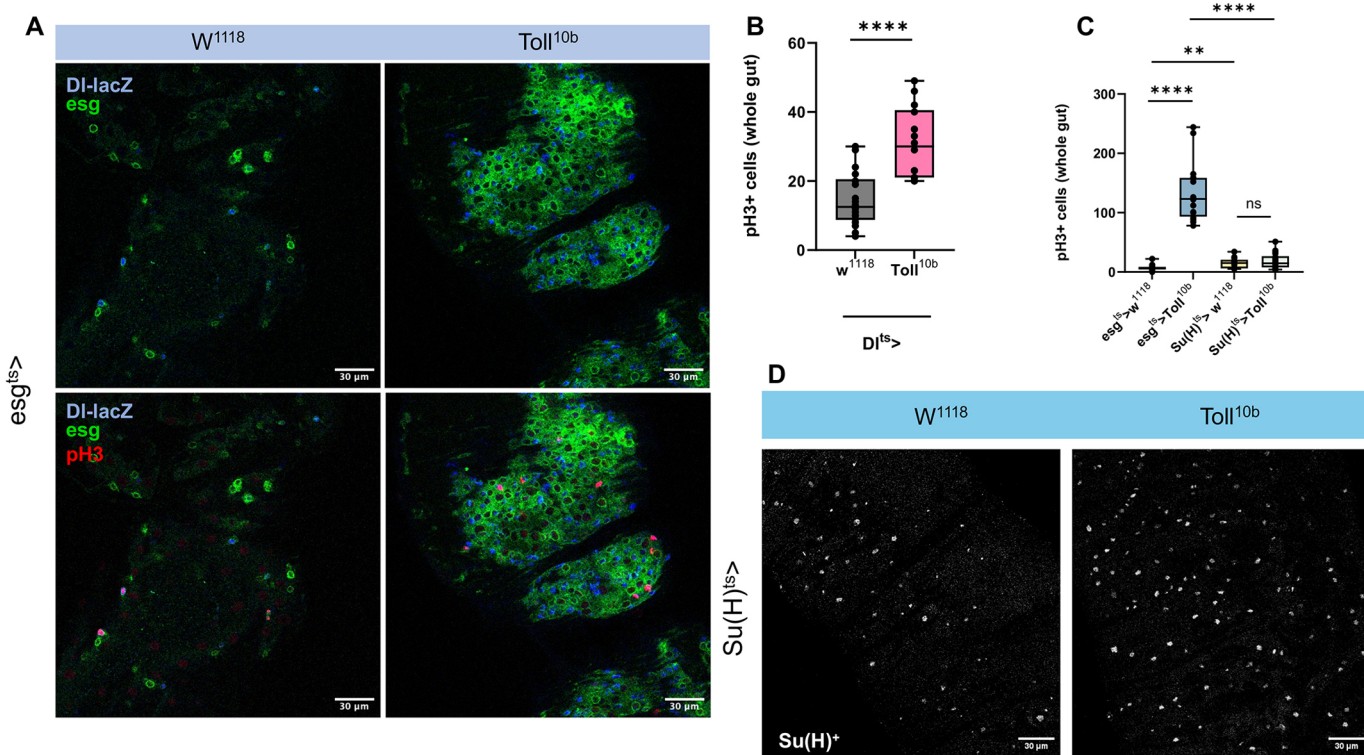

**Fig. 4. Toll activates ISC mitosis and blocks EB differentiation.** (A) Expression of *Toll10b* in progenitor cells increased ISC numbers (*Dl-lacZ*, blue). All mitotic cells (pH3+) coincided with *Dl-lacZ*. (B) Activation of Toll in ISCs significantly increased pH3+ cells (*n*=20). (C) Activation of Toll in EBs did not increase pH3+ cells (*n*=15). (D) Activation of Toll in EBs did increase Su(H)+ cells. All experiments performed with three independent biological repeats. *P*-values were extracted via Mann-Witney test (**P<0.01, ****P<0.0001). Box plots show values of individual flies (dots) with the middle line being the mean and the box indicating the first and third interquartile range. Whiskers show the whole range of values. Scale bars: 30 μm.

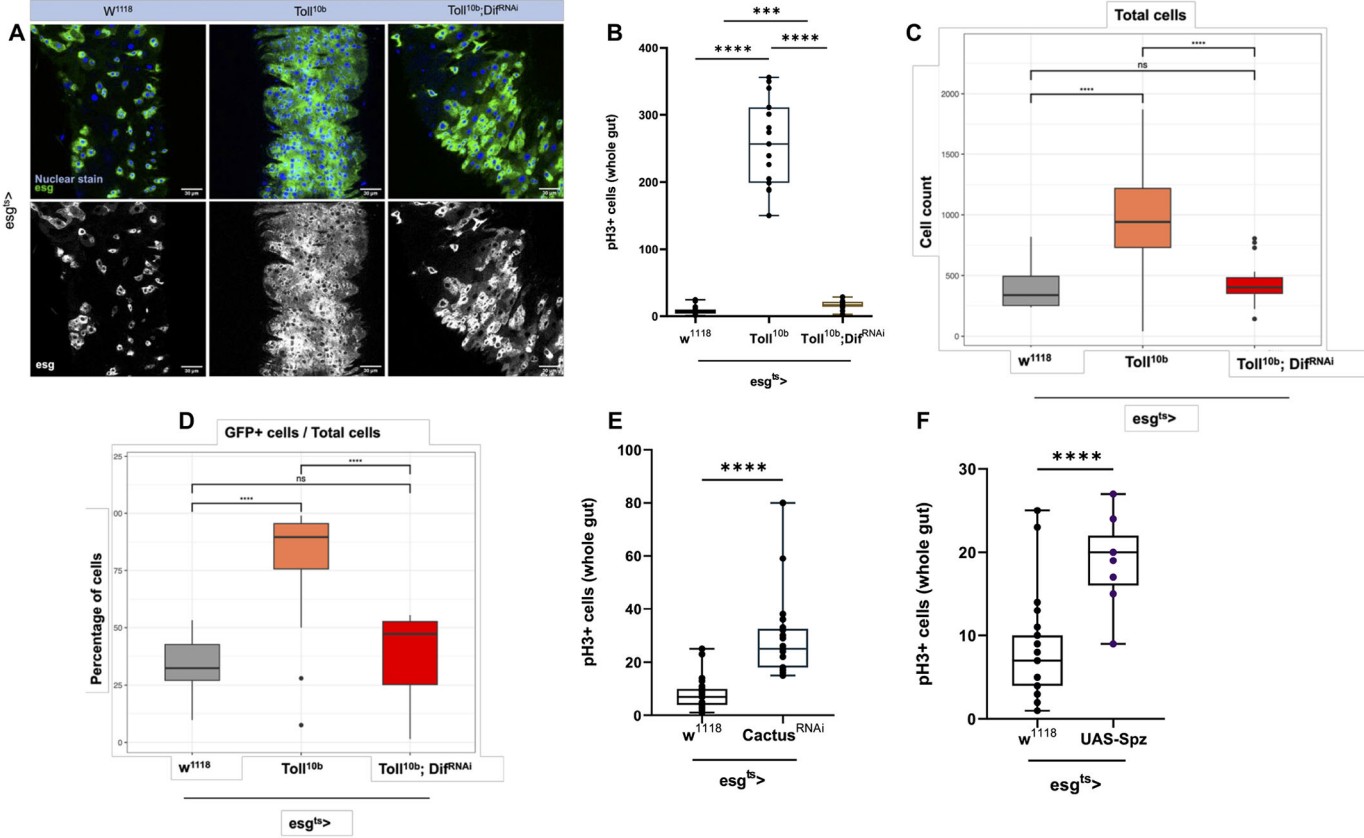

**Fig. 5. The canonical Toll pathway mediates effects of Toll[10b] in the intestinal epithelium.** (A-D) Concomitant *Dif*-RNAi in the context of Toll[10b] significantly reduced progenitor cells (A), intestinal mitosis (pH3[+] cells) (B), total number of cells (C) and the proportion of GFP[+] cells in relation to the total number of cells (D). (E) Knockdown of *cactus* activated the pathway leading to an increase in intestinal mitosis (pH3[+] cells). (F) Overexpression of an activated form of the Toll ligand Spz also increased intestinal mitosis (pH3[+] cells). Statistical comparisons were performed using one-way ANOVA (ns, not significant; ***$P<0.001$, ****$P<0.0001$). Box plots show values of individual flies (dots) with the middle line being the mean and the box indicating the first and third interquartile range. Whiskers show the whole range of values. Both negative ($w^{1118}$) and positive (Toll[10b]) controls in A are the same as in Fig. 3A, as experiments in Figs 3 and 5 were carried out in parallel using the same controls.

$esg^{ts}$-GAL4/UAS-Toll[10b] (Fig. S5B) and $esg^{ts}$-GAL4 control versus $esg^{ts}$-GAL4/UAS-Toll[10b]; UAS-bsk[DN] (Fig. S5C). This suggests that, at least in part, Toll-mediated transcription is potentially facilitated by the JNK pathway. We identified 800 DEGs as unique between control and Toll[10b] flies and 225 as common between Toll[10b] and Toll[10b]; bsk[DN] (Fig. S5D) belonging to a variety of Gene Ontology (GO) categories (Fig. S5E) and cellular pathways as identified by KEGG analysis (Fig. S5F). The top 20 DEGs in the comparison between Toll[10b] and control are shown in Table S1. For the entirety of the results organized in functional categories see Tables S2-S10.

Focusing on the 20 top upregulated genes (excluding Toll, which was as expected the highest expressing gene), we were able to uncover patterns for Toll[10b] activity relative to the control. The MAPK and the JNK pathways were represented with the most highly upregulated targets, namely *p38c*, *drk* and *hep* (JNKK). In addition, a strong target was *dysf*, a transcription factor inducing cell migration by regulating changes in the cytoskeleton (Jiang and Crews, 2006; Rodríguez et al., 2024). Cell proliferation triggered by Toll-dependent transcription was underscored by targets including: *Pvf2*, the VEGF-related secreted factor known for its role in ISC proliferation in an autocrine manner (Bond and Foley, 2012); *upd3*, the cytokine that activates JAK/STAT signalling to trigger ISC proliferation (Zhou et al., 2013); and PI3K21B and Akt, two kinases, the sequential activation of which induces cell proliferation via TOR (Verdu et al., 1999). In addition, *Swim*, encoding a secreted protein that facilitates Wg signalling, was

also a major Toll[10b] upregulated target. Wg is involved in the maintenance and self-renewal of ISCs, and high expression of *Swim* could further enhance such functions as the epithelial surface of ISCs expands (Takashima et al., 2008; Lin et al., 2008). Finally, the most downregulated gene was *multi wing hairs* (*mwh*), a planar cell polarity regulator that inhibits actin polymerization. Its strong transcriptional repression should make actin polymerization available for cell shape changes and movement (e.g. division following mitosis) (Lu et al., 2015). Perhaps unexpectedly, the top downregulated genes included antimicrobial peptides lysozymes (*LysB*, *LysD* and *LysE*, as well as the amidase *PGRP-SC2*). To summarize, the Toll-mediated transcriptional landscape in the gut centres on a group of genes promoting cell proliferation as well as the suppression of a group of antimicrobial genes.

### The JNK pathway is triggered downstream of Toll

The JNK cascade has been implicated in ISC mitosis and epithelial renewal (Biteau et al., 2008). Our transcriptomics data showed that *hep*, a component of the JNK pathway, is a target of Toll. We verified this by measuring *hep* gene expression in a Toll[10b] context (Fig. 6A). Expression of *hep* was also induced by *Ecc15* infection (Fig. 6A). Both cases of *hep* induction were dependent on *Dif* (Fig. 6A). Moreover, increase of ISC mitosis upon Toll activation was blocked when *hep* (JNKK) or *bsk* (JNK), but not *Tak1*, were depleted by RNAi (Fig. 6B). This indicated that the step

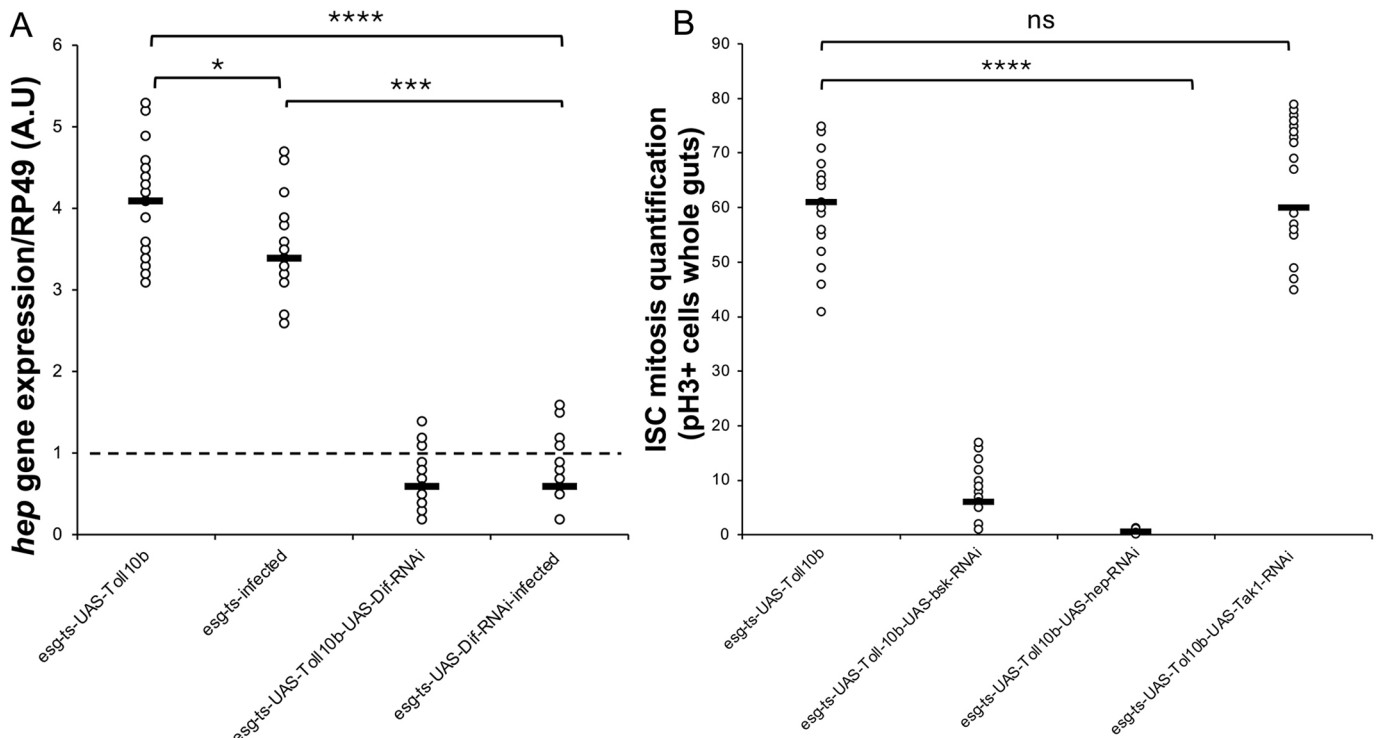

**Fig. 6. JNKK (*hep*) is the point of crosstalk between the Toll and JNK signalling pathways.** (A) Quantitative real time PCR shows that *hemipterous* (*hep*) is activated by both infection and Toll signalling and that this is Dif-dependent. (B) RNAi of *hep* (JNKK) or *bsk* (JNK) but not *Tak1* (upstream of *hep*) blocks ISC mitosis induced by *Toll$^{10b}$*. Statistical significance tests were performed using the Mann–Whitney test (*$P<0.1$, ***$P<0.001$, ****$P<0.0001$). Black line shows mean value. *Hep* expression in *esg$^{ts}$* was placed as 1, while the reference gene was *RP49* (also known as *RpL32*).

in the JNK pathway intersected by Toll signalling was *hep*. To further determine if JNK had any role in the accelerated ISC proliferation observed upon Toll pathway activation, we tested expression of the JNK transcriptional target *puckered* (*puc*) via the *puc-lacZ* reporter (Martín-Blanco et al., 1998) in 10-day *esg$^{ts}$-GAL4* controls and *esg$^{ts}$-GAL4/Toll$^{10b}$* midguts. We observed an increase in *puc-lacZ$^{+}$* cells in *Toll$^{10b}$* flies, an indication of JNK signalling induction upon Toll signalling activation (Fig. S6A). This corroborated the working hypothesis that JNK could potentially be acting downstream of Toll, thereby mediating midgut dysplasia.

We also overexpressed a dominant negative form of JNK (*UAS-bsk$^{DN}$*) in *esg$^{ts}$-GAL4/UAS-Toll$^{10b}$* midguts and assessed whether Toll could still induce the massive expansion of GFP$^{+}$ cells and ISC mitosis. As expected, 10-day-old *esg$^{ts}$-GAL4; UAS-Toll$^{10b}$* midguts were full of GFP$^{+}$ cells (Fig. S6B), displaying a significant increase in the total number of cells (Fig. S6C) as well as the proportion of GFP$^{+}$ in relation to the total number of cells (Fig. S6D) and pH3$^{+}$ cells (Fig. S6E). Strikingly, loss of JNK signalling suppressed Toll-mediated dysplasia in the midgut (Fig. S6B). Moreover, both the total number of cells (Fig. S6C) and the proportion of GFP$^{+}$ cells in the total cell population (Fig. S6D) in *esg$^{ts}$-GAL4/UAS-Toll$^{10b}$; UAS-bsk$^{DN}$* guts were statistically indistinguishable from controls. Finally, reduction in the number of pH3$^{+}$ cells also confirmed that JNK was necessary to mediate Toll-induced expansion of ISCs (Fig. S6E).

Suppression of the above phenotypes was not observed when Wengen, one of the JNK receptors (Fig. S7A), or its ligand Eiger, the *Drosophila* TNF homologue (Fig. S7B), were silenced by RNAi in *esg$^{ts}$-GAL4/UAS-Toll$^{10b}$* guts. Blocking the JNK pathway in homeostatic conditions (*esg$^{ts}$-GAL4, UAS-bsk$^{DN}$*) showed that the number of total cells (Fig. S8A-C) was statistically indistinguishable

from that of the control. On comparing the number of pH3$^{+}$ cells in *esg$^{ts}$-GAL4* controls and *esg$^{ts}$-GAL4; UAS-bsk$^{DN}$* guts, there was a small reduction in pH3$^{+}$ cell numbers ($P<0.05$) (Fig. S8B). However, we interpret the data to indicate that expression of a negative form of JNK in progenitor cells does not significantly influence intestinal epithelial turnover in homeostatic conditions.

Taken together, results in this section provide evidence that JNK is acting downstream of Toll signalling in maintaining epithelial integrity and that the interaction of Toll and JNK signalling is at the level of Hep.

**Akt is important for ISC mitosis downstream of Toll**

An additional Toll target in our transcriptomics was *Akt*, a general signal for cell proliferation (Verdu et al., 1999). Indeed, Toll gain of function (*Toll$^{10b}$*) or *Ecc15* infection induced *Akt* expression (Fig. 7A). This expression was dependent on *Dif*, as Toll activation or infection could not induce *Akt* expression in the context of *Dif*-RNAi (Fig. 7A). Moreover, depleting *Akt* or TOR by RNAi from *Toll$^{10b}$* progenitor cells reduced ISC mitosis to control levels (Fig. 7B). These results indicate that an intact Akt/TOR axis is necessary for Toll$^{10b}$ to exert its effects on ISC mitosis.

**Interaction between Toll and the microbiota at the intestinal interphase**

To test whether this suppression of microbial-regulating factors led to differences in gut bacterial density, we measured the cultivable part of the bacteriome in *esg$^{ts}$-GAL4/UAS-Toll$^{10b}$* flies. In addition, we tested the combination of *esg$^{ts}$-GAL4/UAS-Toll$^{10b}$; UAS-bsk$^{DN}$* to study the interaction of the two pathways in the context of bacterial density regulation. Guts were dissected and plated on MRS agar to determine microbial densities. We found that Toll activation

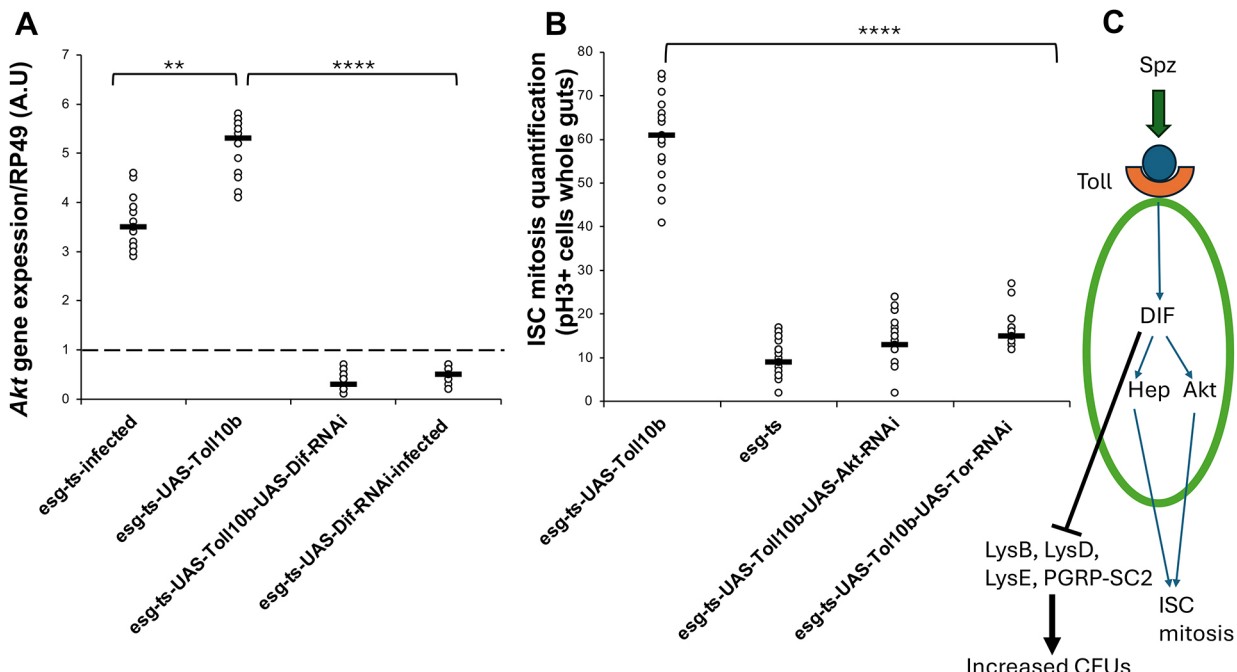

**Fig. 7. Akt is regulated by Toll/Dif.** (A) Quantitative real time PCR shows that *Akt* is activated by both infection and Toll signalling, and this is Dif/NF-κB-dependent. (B) RNAi of *Akt* or *tor* (JNK) block ISC mitosis induced by Toll[10b]. See also Fig S9. Statistical significance tests were performed using the Mann–Whitney test (**P<0.01, ****P<0.0001). *Akt* expression in *esg[ts]* was placed as 1, while the reference gene was *RP49*. Black line shows mean value. (C) Model of how Toll/Dif mediates ISC mitosis and intestinal regeneration through the integration of Hep and Akt activity while increasing gut bacterial density.

significantly increased the cultivable bacterial density (Fig. S9A). This was suppressed by expressing the dominant negative form of JNK (*bsk[DN]*), suggesting that Toll activation needed an intact JNK pathway for the observed increase in colony forming units (CFUs) (Fig. S9A). Increase in bacterial density correlated with the strong transcriptional downregulation of *LysE*, *LysD*, *LysB* and *PGRP-SC2* (see Table S1) as concomitant expression of *UAS-Toll[10b]* with *UAS-PGRP-SC2* or *UAS-LysD* in progenitor cells resulted in suppression of the increase in bacterial density caused by Toll activation (Fig. S9B). However, 16S rRNA library sequencing of gut bacteria revealed that this increase in bacterial density did not affect diversity. At the family level, intestinal bacterial diversity of *esg[ts]-GAL4* control, *esg[ts]-GAL4/UAS-Toll[10b]* and *esg[ts]-GAL4/UAS-Toll[10b]; UAS-bsk[DN]* were indistinguishable (Fig. S9C,D). In the context of *Toll[10b]*, the increase of gut bacterial density did not influence ISC mitosis. The number of pH3[+] cells in germ-free *esg[ts]-GAL4/UAS-Toll[10b]* flies was statistically indistinguishable compared to their conventionally reared siblings and still significantly higher than the germ-free *esg[ts]-GAL4/w[1118]* control (Fig. S9E). This again suggested that increase in ISC mitosis was upstream of (or in parallel to) the increase in gut bacterial density.

## DISCUSSION

As innate immunity bridges the gap between microbes and epithelial cells, it is important to understand the mechanism(s) by which immune signalling interacts with other cellular pathways that maintain epithelial homeostasis. Previous work in *Drosophila* by the Foley lab has identified the IMD/NF-κB/Relish pathway as paramount for the immunological control of ISC survival following infection (Shin et al., 2022; Joly et al., 2025). In the present study, we discover that the Spz/Toll/NF-κB/Dif-Dorsal signalling axis is directly required to promote ISC mitosis, inducing in parallel the increase of gut bacterial density (Fig. 7C).

### Toll integrates JNK and Akt signalling to control ISC mitosis

The interplay of Toll and JNK pathways has been studied in various contexts. In the context of wing and eye development, Toll was found to activate JNK and mediate caspase-dependent cell death through production of reactive oxygen species (Li et al., 2020). The connection of Toll to JNK-mediated cell death in the developing wing has also been documented in cases of chromosomal instability (Liu et al., 2015). In the context of cell competition during wing development, Myc-expressing cells out-compete wild-type cells (a process termed 'super competition', see de la Cova et al., 2014). The 'winner' (Myc[+]) cells produce Spz-Processing Enzyme (SPE) needed to cleave Spz locally, which in turn is required to activate Toll in 'loser' (wild-type) cells (Alpar et al., 2018). Activation of Toll/NF-κB signalling in 'loser' cells triggers their apoptosis and eliminates them (Alpar et al., 2018). Finally, Spz-Toll-mediated cell death through JNK signalling has also been observed in larval epithelial tumours, where Toll activity restricts tumour growth (Parisi et al., 2014).

In contrast to the above and in the context of the adult gut, JNK was found to induce ISC proliferation (Perdigoto et al., 2011). Sustained JNK activity led to mis-differentiated daughter cells that expressed *esg* but were polyploid (characteristic of an EC) while not expressing EC-specific markers (Perdigoto et al., 2011). Our data indicated that activation of Toll in progenitor cells also led to accumulation of cells, that had larger nuclei as compared to ISC/EB but were GFP[+], while *hep* was a Toll transcriptional target. Hence, we explored the interaction of JNK and Toll pathways. We found that, in progenitor cells with activated Toll signalling (*Toll[10b]*), RNAi of *hep* (JNKK) or *bsk* (JNK), but not *Tak1*, *egr* or *wgn*, blocked ISC mitosis. Moreover, when we expressed *bsk[DN]*, a non-activable form of JNK, in *Toll[10b]* progenitors, we rescued intestinal dysplasia, bringing down the levels of total cell numbers, GFP[+] progenitor cell proportions and pH3[+] cell counts to the levels of

controls. This clearly implied that Toll and JNK signalling intersected at the level of Hep in inducing ISC proliferation. Consistent with previous studies, we have determined that Akt signalling is involved in ISC proliferation (Foronda et al., 2014; Mattila et al., 2018). Here, however, we further pinpoint that Toll/Dif/NF-κB signalling is the upstream control module of *Akt* gene expression and the role of Akt in ISC mitosis requires an intact Toll pathway (Fig. 7C).

### Toll activity controls the intestinal ecosystem

$Toll^{10b}$ activation of ISC mitosis and accumulation of intestinal progenitors was accompanied by a significant increase of cultivable gut bacterial density. At first glance this appeared to be counterintuitive. How could an anti-microbial response increase gut bacterial density? However, looking at the Toll transcriptional signature no AMPs were upregulated, while genes coding for enzymes such as lysozymes and peptidoglycan amidases were strongly downregulated. On measuring cultivable bacterial colonies in the gut of control, $Toll^{10b}$ and $Toll^{10b}$; $bsk^{DN}$ flies, we found that blocking JNK also suppressed Toll-mediated increase of the cultivable bacterial density in the gut. The increase in bacterial density upon Toll activation did not stimulate ISC mitosis further. This was evident as there was no reduction of pH3+ cells in $Toll^{10b}$ germ-free flies.

Spz is necessary for midgut mitosis in progenitors (specifically ISCs) under both homeostatic and infected conditions. Upon *P. aeruginosa* infection, *spz* is necessary in progenitor cells, ECs, EEs (to a lesser extend), the trachea and the fat body. We speculate that since *spz* is a Toll target (De Gregorio et al., 2002) and there are several sources for it (both in and away from the gut), there may be a paracrine signal sent to ECs. We have previously shown that Toll activation in ECs triggers lipid catabolism, which is important for the maintenance of gut bacteria (Bahuguna et al., 2022). An increase in the supply of Spz to ECs, may lead to a rise in lipid catabolism and, therefore, in triglycerides available to bacteria, leading to density increase. This should be combined with the suppression of lysozyme and amidase gene expression when Toll is activated that also leads to increase in commensals. Alternatively, the cellular mass increase in $Toll^{10b}$ flies could slow gut motility and thus increase bacteria numbers in transit. Functional gastrointestinal disorders in humans and mice have shown that gut bacterial density is connected to intestinal motility (reviewed by Sanders et al., 2012). More work is needed to investigate the validity of the above working models.

### Conclusion

Our data indicate that Toll is important for regeneration of the intestinal epithelium, but its activation does not lead to the 'classical' antimicrobial immune defence transcriptional induction. Instead, Toll signalling is necessary and sufficient for ISC mitosis through integrating JNK and Akt signalling, as well as for increasing the density of gut bacteria to counter the effects of an infection.

## MATERIALS AND METHODS
### *Drosophila* stocks
Stocks used were: $w^{1118}$ [Bloomington *Drosophila* Stock Center (BDSC), 3605]; $esg^{ts}$ (Buchon et al., 2013); $ISC^{ts}$ and $esg^{ts}$; *Dl-lacZ* were both kind gifts from Bruce Edgar (University of Utah, USA); $UAS-Toll^{10b}$(II) (Shia et al., 2009); $UAS-Toll^{10b}$(X) (BDSC, 58987); $UAS-Cactus^{RNAi}$ (BDSC, 31713); $UAS-Dif^{RNAi\_1}$ (BDSC, 30513); $UAS-Dif^{RNA\_2}$ (BDSC, 29514); $UAS-Dif^{RNAi\_3}$ [Vienna *Drosophila* Resource Center (VDRC), 100537]; $UAS-Bsk^{DN}$ (BDSC, 6409); $UAS-spz^{GFP}$ (Cho et al., 2010); *puc-lacZ* (Martín-Blanco et al., 1998); $UAS-tak1^{RNAi}$ (VDRC, 101357); $UAS-dorsal^{RNAi}$ (VDRC, 105491); $UAS-Toll^{RNAi}$ (VDRC, 100078); $UAS-spz^{RNAi}$

[BDSC, 28538 (line 1) and VDRC, 105017 (line 2)]; *Toll MiMIC* (BDSC, 36134); *spz MiMIC* (BDSC, 34313); *Dif MiMIC* (BDSC, 33078); *Dorsal MiMIC* (BDSC, 37091).

### Infection model
To prepare the bacterial culture, it was grown in a flask containing Rifampicin in 1× LB broth. Optical density (OD) at 600 nm was measured using a nanodrop. Mated female flies were used for these experiments.

For PA14 infection, PA14 feeding was performed as previously described (Apidianakis and Rahme, 2009). In short, a single colony of *P. aeruginosa* strain PA14 was grown at 37°C in liquid LB broth, to $OD_{600nm}$=3, corresponding to $5\times10^9$ bacteria/ml. Female mature flies of the desired genotype were starved for 5 h and added in groups of ten per fly vial containing a cotton ball at the bottom soaked with 5 ml of 0.5 ml PA14 $OD_{600nm}$=3, 1 ml 20% sucrose and 3.5 ml $ddH_2O$. For uninfected control, 1 ml sucrose 20% and 4 ml $ddH_2O$ was used. Flies were incubated for 48 h at 29°C (for all experiments using the GAL4-UAS, unless otherwise noted). Female mated 5- to 7-day-old flies were used for all feeding assays.

### Polylysine slide preparation
For preparation of the polylysine coated slides, the contents of a 1 g bottle of polylysine (Sigma-Aldrich, P1524) were dissolved in 170 µl $ddH_2O$ in a small beaker using a magnetic flea stirrer. After the solution appeared clear, 10 ml of this solution was aliquoted and stored at −20°C as 10× polylysine stocks. When the slides needed coating, three aliquots of the stock solution were defrosted and used. Working solution was made by mixing 30 ml of 10× polylysine solution with 170 µl $ddH_2O$ and 1.5 ml of Kodak Photo-Flo. The glass slides were loaded onto a staining jar and the working solution was poured into the jar. The slides were left in polylysine solution on the rocker for 10 min. The solution was then poured out into a falcon tube and the slides were dried at 60°C for 10 min in the oven. The slides underwent a total of five polylysine coating and drying cycles. The slides were then air-dried. After drying, the surface of the slide that had the polylysine coat was bordered with silicone to create a well to perform immunostaining steps and hold antibody solutions. When the silicone ring was dry, the slides were stored at 4°C.

### Gut dissection and immunostaining
The polylysine coated slides make the *Drosophila* gut stick to its surface and have been used for all the steps of immunostaining to preserve the integrity of the whole length of the tissue. For gut immunostaining, individual guts from anaesthetised flies were dissected at selected time points (10 days adult flies) (*n*=10-15) into cold 1× phosphate buffered saline (PBS). The guts were then arranged onto the prepared polylysine slides containing cold 1× PBS in the well, made of silicone. After arranging the samples on the slide, the PBS was replaced with 4% formaldehyde fixative diluted in 1× PBS for 40 min. The slide was always covered so the samples remained in the dark in all the steps moving forward. After fixing, guts were washed twice in 0.1% 1× PBS and 0.1% Triton X-100 (PBST) for 5 min per wash cycle. The sample was then incubated in 1% bovine serum albumin (BSA) diluted in 0.1% PBST for 50 min (blocking). Guts were then incubated in the primary antibody of desired dilution overnight at 4°C. The primary antibody was then removed and the guts underwent three washing cycles with 0.1% PBST for 5, 20 and 10 min per cycle, in that respective order. The guts were then incubated in secondary antibody of desired concentration for 1.5 h at room temperature. The guts were then washed with 0.1% PBST for 10, 20 and 15 min per cycle. After removing the PBST, the guts were incubated in 1:1000 concentration (diluted in PBS) of nuclear stain To-Pro-3 (Invitrogen-T3605) for 25 min. After two more rounds of wash cycles for 5 min each, VECTASHIELD (anti-fade mounting medium H-1000-without DAPI, Vector Laboratories) was added to the sample and the slide was covered with a coverslip. The coverslip was then sealed using nail varnish. The slides were stored at 4°C and imaged within 3-4 days. The control guts were placed on the same slide as experimental guts for direct comparison.

The following primary antibodies were used at specified concentrations and diluted using 0.1% PBST in 1% BSA: rabbit Phospho-Histone H3 (Ser10) polyclonal antibody (PA5-17869, Invitrogen, 1:200); mouse Prospero (MR1A) (AB_528440, Developmental Studies Hybridoma

Bank, 1:50); anti-β galactosidase mAb (Z3781, Promega, 1:500); anti-Dif (Ligoxygakis et al., 2002). The secondary antibodies used were: goat anti-rabbit IgG (H+L) highly cross-adsorbed secondary antibody, Alexa Fluor™ 568 (A-11036, Invitrogen, 1:200) and Alexa Fluor 546 anti-mouse (A-21123, Invitrogen, 1:200). Experiments presented in Figs 1 and 2 used the immunostaining protocol and reagents described in Apidianakis and Rahme (2009).

### Imaging

The samples were imaged using a Leica TCS SP5 confocal laser scanning microscope and all images were captured using 63× magnification objective with a numerical aperture (NA) of 1.4. The oil used for microscopy was Leica Type F Immersion liquid ($n_e^{23}$=1.5180 and $v_e$=46). The Leica SP detector was used for capturing images. The voxel size of the images was (x;y;z)=481.47 nm; 481.47 nm; 2 μm.

Z-stacks of each gut was taken at the R4 region of the posterior midgut. Every sample varied in thickness, due to variation in gut physiology or due to slight variations in sample preparation. To have uniform images and consistent sampling of the guts, z-stacks were taken from the start of the gut to the middle of the gut, ending at the lumen. The midpoint was calculated by dividing the start and the end by two. Imaging was only done till the mid-point to avoid deteriorated resolution and quality of images with increased penetration into the sample depth. We took 2 μm thick slices for the z-stack (Fig. S10).

The samples used in this project had progenitor cells labelled with GFP. Therefore, all samples had a channel for the detection of GFP which was kept consistent through all the samples. For imaging GFP in guts, Argon laser was powered up to 30% of its maximum capacity. The excitation wavelength used was 488 operating at 20% of the maximum intensity. The gain was set to 910. The emission wavelength was 502-568 nm. To-pro-3 used to label the nucleus was a far red stain. The excitation wavelength used for its detection was 633 nm operated at 10% of the maximum intensity with the gain in the range of 840-850. For detection of Alexa Fluor 568 secondary antibody the excitation laser used was 543 nm and emission was set at 565-635 nm. For detection of the proteins probed with Alexa Fluor 546 antibody, the excitation wavelength used was 543 nm and the detection wavelength used was 554-635 nm. The excitation laser intensity and gains were varied to account for the abundance and localization of the protein of interest targeted with the antibody. The images were visualized using Fiji-2 (ImageJ).

### Image processing and analysis

To measure nuclear size, nuclei channels were first median filtered using the Scipy 'ndimage' module (Virtanen et al., 2020) at a kernel radius size of 3 px. Filtered images were then segmented using Cellpose v2.0.1 (Stringer et al., 2021) with 3D mode enabled. Cell size assessment was made with major axis length of each nuclei quantified using Scikit-Image 'regionprops' function (van der Walt et al., 2014). The large and small nuclei cells were filtered based on a threshold set from histograms plotted for density of cells against their major axis length for control fly guts, and this threshold was used for all other samples (Fig. S11).

### Quantification of fluorescence intensity levels

Segmented nuclei masks were expanded by 2 px in 3D and the original nuclei masks were subtracted to obtain a 3D ring mask around the nucleus, representing a sample of cytoplasmic volume. Fluorescence channels to be quantified were background subtracted (rolling ball, 20 px radius) and were quantified within 3D ring masks using Scikit-Image 'regionprops' function. Intensity thresholds were set by plotting a distribution of integrated fluorescence intensities and by setting 1.5× normalized fluorescence cut-off from internal control groups. Statistical comparisons were performed in the R (4.0.4) environment. All the statistical tests used have been described in the figure legends. All analysis scripts are available at https://github.com/jefflee1103/Image-analysis/tree/main/people/Ash_U.

### Quantification of pH3⁺ cell numbers

For quantifying the pH3⁺ cell counts for the whole gut, argon laser arc lamp was used. This allowed for visualizing pH3⁺ cells through the eyepiece of the microscope using an excitation filter of 515-560 nm and emission filter of 590 nm. pH3⁺ cells were counted manually, going through the anteroposterior axis of the gut stack by stack using the eyepiece. Approximately 10-15 guts from three different crosses were used to quantify pH3⁺ cell counts.

### RT qPCR

RT-qPCR was performed as described previously (Tamamouna et al., 2021). For Fig. S3, total RNA was prepared from L3 larvae (three larvae per sample, three biological replicates per genotype). Expression of spz was normalized to the expression levels of two reference genes, RpL32 and Gapdh1, using the $2^{-\Delta\Delta Ct}$ method using the BioRad CFX Manager 3.1 software. The primer sequences were: spzL 5′-GTGATTCTGGAA-AATGGGATTC-3′ and spzR 5′-TCTGTGGTGGGTGAAACTTCT-3′; rpl32L 5′-CGGATCGATATGCTAAGCTGT-3′ and rpl32R 5′-CGACG-CACTCTGTTGTCG-3′; gapdh1L 5′-GCTCCGGGAAAAGGAAAA-3′ and gapdh1R 5′-TCCGTTAATTCCGATCTTCG-3′.

### RNA-sequencing

Sequencing libraries were generated using the MGIEasy RNA-seq Library Preparation Universal Kit by BGI Genomics, Hong Kong. RNA sequencing was performed on the DNBSEQ-G400 using the 150 paired-end sequencing protocol. Three samples ($esg^{ts}/w^{1118}$, $esg^{ts}/UAS$-$Toll^{10b}$, $esg^{ts}/UAS$-$Toll^{10b}/UAS$-$Bsk^{DN}$) with three independent biological repeats were used.

### Culture-dependent quantification of gut microbiota

Five Drosophila guts were dissected in 1× PBS solution and then homogenized using a sterile needle homogeniser until the guts appeared to be completely broken. The resulting suspension was then diluted to form 1:100 dilution and 200 μl was plated onto three plates containing de Man, Rogosa and Sharpe (MRS) agar and incubated at 30°C for 48 h. The colonies in each plate were counted, and $\log_{10}$ values of the CFUs were calculated. These three plates formed the technical replicates for the experiment. This entire procedure was repeated thrice using flies from different generations to account for biological variations in samples. The colonies were counted using ImageJ by setting thresholds to form masks containing the colonies and then using the particle counter function. The $\log_{10}$ CFU values of mutants and controls were statistically compared using unpaired, two-tailed Student's t-test, and graphs were plotted in GraphPad Prism.

### Culture-independent quantification of gut microbiota

Identification of microbial families by analysing their 16 s ribosomal RNA gene sequences using the hypervariable region V3 to identify microbial composition of the Drosophila gut was performed as previously described (Bahuguna et al., 2022).

### Acknowledgements

We thank the Bloomington Drosophila Stock Centre and the Vienna Drosophila Resource Centre for fly stocks, BGI Genomics, Hong Kong, for RNA sequencing, as well as Alexandros Chrysanthou and Vasilia Tamamouna for help with experimental work.

### Competing interests

The authors declare no competing or financial interests.

### Author contributions

Conceptualization: C.P., P.L.; Data curation: A.U., S.B., Y.A., C.P., P.L.; Formal analysis: A.U., F.S., G.P., C.M., Y.A., C.P., P.L.; Funding acquisition: Q.X., Y.X., M.B., J.Z., C.P., P.L.; Investigation: A.U., F.S., T.H., G.P., S.B., C.M., J.Y.L., Q.X., J.Z.; Methodology: J.Y.L., J.Z., Y.A., C.P., P.L.; Project administration: P.L.; Supervision: M.B., C.P., P.L.; Writing – original draft: A.U., P.L.; Writing – review & editing: P.L.

### Funding

P.L. was funded by the EPA Cephalosporin Fund (CF 388) and the John Fell Fund, University of Oxford (0010611), as well as the Changjiang Scholar Program, Ministry of Education as a visiting Professor at Chongqing University. C.P. and Y.A. were supported by the University of Cyprus. J.Z. is supported by Startup funding from Hunan University, National Natural Science Foundation of China (32270890) and the Department of Science and Technology of Hunan Province (2023JJ0007). Open Access funding provided by the University of Oxford. Deposited in PMC for immediate release.

### Data and resource availability

RNA-sequencing data have been deposited in Gene Expression Omnibus under accession number GSE315850. All analysis scripts are available at

**Peer review history**

The peer review history is available online at https://journals.biologists.com/dev/lookup/doi/10.1242/dev.204794.reviewer-comments.pdf

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
