## [Peer Review File · Development (Cambridge, England)]

Toll signalling controls intestinal regeneration in *Drosophila*

Aiswarya Udayakumar, Filippou Stavropoulos, Theodosia Hadjipanteli, Guofan Peng, Shivohum Bahuguna, Caitlin MacClay, Jeffrey Y. Lee, Qi Xiao, Yuxian Xia, Michael Boutros, Jun Zhou, Yiorgos Apidianakis, Chrysoula Pitsouli and Petros Ligoxygakis

DOI: 10.1242/dev.204794

Editor: Irene Miguel-Aliaga

Review timeline

Original submission:	17 March 2025
Editorial decision:	22 April 2025
First revision received:	19 September 2025
Editorial decision:	13 October 2025
Second revision received:	12 November 2025
Accepted:	20 November 2025

Original submission

First decision letter

MS ID#: dev.204794

MS Title: Toll signalling controls intestinal regeneration in *Drosophila*

Authors: Aiswarya Udayakumar; Filippou Stavropoulos; Theodosia Hadjipanteli; Guofan Peng; Shivohum Bahuguna; Caitlin MacClay; Jeffrey Y. Lee; Qi Xiao; Yuxian Xia; Michael Boutros; Jun Zhou; Yiorgos Apidianakis; Chrysoula Pitsouli; Petros Ligoxygakis

Article Type: Research Article

Dear Dr Ligoxygakis,

I have now received all the referees' reports on the above manuscript, and have reached a decision. The referees' comments are appended below, or you can access them online: please go to:

As you will see, the referees express considerable interest in your work, but have some significant criticisms and recommend a substantial revision of your manuscript before we can consider publication. If you are able to revise the manuscript along the lines suggested, which may involve further experiments, I will be happy receive a revised version of the manuscript. Your revised paper will be re-reviewed by one or more of the original referees, and acceptance of your manuscript will depend on your addressing satisfactorily the reviewers' major concerns. Please also note that Development will normally permit only one round of major revision. If it would be helpful, you are welcome to contact us to discuss your revision in greater detail. Please send us a point-by-point response indicating your plans for addressing the referees' comments, and we will look over this and provide further guidance.

Please attend to all of the reviewers' comments and ensure that you clearly highlight all changes made in the revised manuscript. Please avoid using 'Tracked changes' in Word files as these are lost in PDF conversion. I should be grateful if you would also provide a point-by-point response detailing how you have dealt with the points raised by the reviewers in the 'Response to Reviewers' box. If you do not agree with any of their criticisms or suggestions please explain clearly why this is so.

Reviewer 1

In this paper by Udayakumar et al., the authors report a non-immune function of Spz/Toll signaling in intestinal stem cell proliferation during pathogen induced regeneration of the adult fly midgut. Toll signaling activation is necessary and sufficient to drive intestinal stem cell proliferation in a mechanism that involves JNK/AKT/mTOR pathway activation and the important contribution of the signaling ligand Spz from different cellular compartments in the midgut.

Given that IMD has been classically thought to mediate immune signaling in the gut, this represents an intriguing role of Toll signaling in adult tissue homeostasis in *Drosophila*. Several open questions remain to be addressed to consolidate the work and main conclusions of this study.

Major comments

- 1- Are functional roles of Toll signaling limited to regeneration in response to pathogenic infection? Other damaging agents (e.g. DSS, Bleomycin) should be tested.
- 2- Is there an effect from knocking down Toll signaling in EB cells only?
- 3- Is expression of the ligand Spz regulated by damage? Which cells produce this ligand? Functional experiments have been conducted using RNAi knockdown in different cell types. However, the use of antibodies or reporter lines would be important to localize the source of the ligand.
- 4- How is Toll signaling activated in response to damage? In other words, what lies upstream of Spz/Toll activation in regeneration?
- 5- Is there a role for Spz/Toll signaling in intestinal homeostasis? This has been tested through the use of *esg>toll* overexpression tools. However, clonal analysis over time, using gene loss of function should be done to address this question. Using *esg* flip out or MARCM system and staining with differentiation markers would be important.
- 6- The same applies to the data in Figure S10. Clonal analysis needs to be used in such setting.
- 7- Related to the above point, the effect of Toll overexpression in cell differentiation should be clarified. In Figure S5 the authors show that overexpression of Toll signaling under the control of *esg-gal4* leads to increase in EE cells. On the other hand, they show that overexpression of Toll under *Su(H)-gal4* (EB specific *gal4*) leads to increase in EB cells (Figure 3B). This is not explored further. Is there a relationship between the EE and EB phenotype presented? Does gene loss of function in the same cell types produced the opposite result (in control or infected midguts)?
- 8- Figure S2B: The expression pattern of *Dif* does not appear to overlap with *esg>RFP* cells but rather be complementary to it.
- 9- What are the biological implications of reducing toll signaling in the intestine. Analysis of organismal health and/or lifespan in homeostatic or infective settings would be important.

Minor comments

- 1- There are several graphs with missing data (Figures 1B, S1A, D).

Reviewer 2

In their paper, "Toll signaling controls intestinal regeneration in *Drosophila*" the authors provide fairly convincing evidence that activation of the Toll pathway is necessary and sufficient for ISC mitosis and that JNK and AKT signaling are downstream of Toll activation. As the authors point out, the role of the Toll pathway in the fly midgut has been understudied. The author's work is an important start to establishing a role for the pathway. However, as it stands, I have some major concerns.

- 1) The overreliance on RNAi knockdown using *esg-Gal4* makes it difficult to determine what the phenotypes are exactly. A lot of the phenotypes seen could result from, as the authors put it, "massive" increases in cells and subsequent induction of cell death and injury. Rare, marked clones, as suggested below, would help to obtain more precise conclusions about the role of Toll signaling.

- 2) It is unclear to me, after reading this paper, what the purpose of Toll signaling is. The Discussion section is mostly a rehash of the results. It would help to have some insight into why there is a need for this pathway to drive cell division when there are many others that can do this.

SUGGESTIONS TO AUTHORS

Introduction:

- 1) Line 46: "was driving" should be "drives"
- 2) The authors should strive to cite more primary papers in the manuscript. While it is okay to include reviews, it makes it difficult for readers to determine which primary papers established the points the authors are trying to make.
- 3) Line 93: Micchelli and N. Perrimon 2006 does NOT use Delta to identify ISCs.
- 4) Line 103: Ohlstein and Spradling 2006; Ohlstein and Spradling 2007 do not mention *esg*. Better to cite: Micchelli and N. Perrimon 2006
- 5) Line 110-111. "as the posterior midgut is more immunologically active." My impression is that bacteria are found in the anterior. Could the authors provide evidence for the posterior midgut being more active? Also, in the paper the authors characterize the whole gut.

Results section:

- 6) The authors should list references for the genetic lines they use in the results section.
- 7) *Myo1A* becomes expressed in ISC with age.
- 8) Figure 1B. The data (bars) are missing from the graph. Same with Figure S1 and S1D.
- 9) I'm surprised there are any mitosis in flies reared on sucrose, which lacks protein.
- 10) The *Dif* RNAi result does not look that different from the control in figure 1. Did the authors validate the line?
- 11) It would help the author's case if the authors made marked clones of Toll pathway mutants, both LOF and GOF. This would allow them to use validated alleles (nulls and hypomorphs) and to examine clone growth and composition. This would give definitive data regarding division rates and cell fate choices. I would suggest they do this with *UAS-spz* as well.
- 12) "Figure S1. Toll is not required in mature intestinal cells for midgut mitosis" This is not an appropriate conclusion. First of all, the RNAi may not come on until after the EC has been determined. Second of all, the authors provide no firm evidence that the protein activity of these genes are affected by RNAi expression. They should just temper their statement.
- 13) The authors should be cautious with conclusions using ISCs. The RNAi expressed in the ISC might be inherited by the EB and cause its effects there.
- 14) Figure S2B. What is the cell with green *Dif* staining but no mCherry (Left arrow) staining? How many cells become positive after infection? A low mag view would be helpful. Do the authors have any insight into which progenitor is positive for *Dif*? The ISC or the EB?
- 15) Line 173. "into its native locus (Fig 2C)". I think they mean Figure S2C.
- 16) Figure S3 is confusing. If ISCs is a marker of ISCs then it should be roughly evenly distributed throughout the intestine in *w1118*. The data makes me think that it might mark a subset of ISCs, in some state of the cell cycle. Perhaps ones that are getting ready to divide. This would make sense, given that the *Toll10B* and *UAS-DIF* flies have more PH3 and more GFP cells. Otherwise, I would have to conclude *spz*, *Dif*, and *PGRP* RNAi flies have almost no stem cells. If so, the authors need to do an appropriate characterization of this phenotype. Are ISCs dying or differentiating? In any event, the authors need to clarify this.
- 17) It would be helpful to demonstrate that *Dif* protein is localized to the nucleus in all progenitors in *esg>UAS-Toll10B*.
- 18) I worry that constitutive activation of the Toll pathway could create artifacts. The authors should look at PH3 after heat induction of *UAS-spz*.
- 19) The word "massive" is vague.
- 20) It's unclear to me what the *esg>GFP* cells are in *Toll10B*. *esg* can mark enterocytes with stress, damage, and aging. Staining with *Pdm1* could help with this. Also, as mentioned above, marked clones expressing *spz* or *Toll10B* would make it easier to assess the nature of the cells.
- 21) Line 219: (Ohlstein and Spradling, 2007) did not use *DI-lacZ*. I would also caution using this reporter as it has been shown that *DI* can be expressed in non-ISCs with injury. Also, work from the Jiang lab suggests EBs can undergo mitosis.
- 22) "The legend of Figure 3D and the text states "Activation of Toll in EBs did increase *Su(H)*+ cells." Where is the quantification for this conclusion?
- 23) Figure S6A (*Toll10B*, *Dif* RNAi) is not consistent with the quantification of the data in the C and D. Looks like the decrease in GFP cells is partial compared to *w1118*.

- 24) Figure S7A. A region appears negative for GFP. What region do the authors think this is?
 25) What was the rationale to look at JNK signaling by RNA-seq?
 26) Line 324-325: "These data indicate that JNK signaling is not necessary for the maintenance of epithelial turnover in homeostatic conditions." It's more accurate to say it is not necessary for cell division in homeostatic conditions." Has this not been addressed by the Jasper lab?
 27) What do the authors mean by "epithelial integrity" in line 328?

Reviewer 3

SUMMARY OF THE ADVANCE MADE IN THIS PAPER AND ITS POTENTIAL SIGNIFICANCE TO THE FIELD

Udayakumar et al. undertook the advantage of *Drosophila* genetic tractability to show in detail and confirm consistently that the Spz/Toll/NF- κ B/Dif signalling axis is required to promote Intestinal Stem Cells (ISC) mitosis. In addition, they connect Toll activation with the increase of gut bacterial density. They further show Toll activity was mediated by JNK and Akt/mTOR signalling, giving more insights into ISC regulation. The Toll control of intestinal homeostasis is an important discovery of general interest. Overall, the work is well-performed and includes a variety of experimental approaches. However, there are certain concerns that remain to be addressed.

SUGGESTIONS TO AUTHORS

- This work nicely demonstrates the role of Toll pathway in the control of stem cell proliferation under homeostasis and infection. However, an important question that remains to be addressed is: what is the relevance of the identified mechanism? How will knockdown of Toll and subsequent epithelial regeneration affect fly survival under homeostatic conditions (and aging), and also will they survive infection differently in comparison to controls? In another words, will impaired epithelial renewal due to Toll pathway suppression make flies more susceptible to infections?
- It would be useful to include as supplementary table the full list of DEGs and not only the top 20.
- Regarding the presentation of data, I would ask the authors to include missing box plot data from Figure 1B, Figure S1A, and Figure S1D, and for those graphs, do the statistical comparison of P.a-treated flies vs sucrose-treated flies. Otherwise, the separation of data to analyze the comparison of RNAi to control flies within the same treatment, like in Figure 1 and Figure S1, is redundant.
- Regarding the statistical analysis and interpretation of data, here are my specific observations and concerns:
 - Currently, if understood correctly, Figure 1B should have the same data and analysis as Figure 1C and 1D - so double check whether spz RNAi suc is * or ** significant.
 - Line 322 In 'and the proportion of GFP+ cells (Fig. S10B) were statistically indistinguishable from that of the control. On comparing the number of p3+ cells in esgts-GAL4 controls and esgts-GAL4; UAS-bskDN guts, there was a small reduction in p3+ cell numbers ($p < 0.05$)' - Please, exclude or clarify whether Fig. S10B was statistically indistinguishable from that of the control, since the plot shows significant differences.
 - Figure 5B ESG-ts is a control, so it should go first, and asterisk lines should be clarified (why is it only one long line?).
 - The authors are advised to make their graphs consistent, for example Fig. 2B is different vs 2C-F. Why not to use uniform graph format?
 - Figure S2A should be improved. First, as I understand each genotype has uninfected and infected conditions - those should be clearly marked on the graph. Second, the aim of this graph is to compare Toll RNAi treatments with w1118 after Ecc15 infection. Hence, these comparisons should be shown on the graph. Third, include the driver label (ISCts) on the plot.
- To obtain clarity and easier reading for wider audience, I recommend including the explanation of used fly lines when they are first mentioned in the text. Examples:
 - EE-GAL4 is mentioned in the results (Line 150) but was not included when mentioning all GAL4 lines (Line 139).
 - esgts-GAL4/UAS-Toll10b; UAS-bskDN is first mentioned and used in the experiment in Line 259, but explained only later in text in Line 307 (...also overexpressed a dominant negative form of JNK (UAS-basketDN, or UAS-bskDN...)).
 - Elaborate the usage/mentioning of tak1 (Line 298).

In addition, for clarity in the methods part, either put PA14 protocol before Ecc15, or already mention that female mated flies were used for all experiments.

6. Again, to obtain clarity and easier reading for the wider audience, I would recommend including a schematic of gut cells (something like: https://www.researchgate.net/figure/Proliferation-and-differentiation-of-intestinal-stem-cell-ISC-in-Drosophila-The_fig1_376715092) and inclusion of GAL4 lines that you used to test them in this paper.

Specific recommendations:

Figure 6B include the driver label on the plot and figure description

Line 451 include the Rifampicin concentration

Line 573 write the volume of PBS that was used

Line 66 D-V add 'dorsoventral' before abbreviation

Line 188 not so relevant here: Drosophila ISC numbers increase with age leading to dysplasia in older guts - more connected later to Line 257

First revision

Author response to reviewers' comments

Reviewer 1:

In this paper by Udayakumar et al., the authors report a non-immune function of Spz/Toll signalling in intestinal stem cell proliferation during pathogen induced regeneration of the adult fly midgut. Toll signalling activation is necessary and sufficient to drives intestinal stem cell proliferation in a mechanism that involves JNK/AKT/mTOR pathway activation and the apartment contribution of the signalling ligand Spz from different cellular compartments in the midgut.

Given that IMD has been classically thought to mediate immune signalling in the gut, this represents an intriguing role of Toll signalling in adult tissue homeostasis in Drosophila. Several open questions remain to be addressed to consolidate the work and main conclusions of this study.

Major comments

1- Are functional roles of Toll signalling limited to regeneration in response to pathogenic infection? Other damaging agents (e.g. DSS, Bleomycin) should be tested.

Response: We have tested regeneration in response to damage (after DSS treatment) in new experiments (see new Fig. S1D). [Lines 193-196].

2- Is there an effect from knocking down Toll signalling in EB cells only?

Response: Yes, there is. EB-specific silencing of core components of the Toll pathway reduces midgut mitosis upon infection albeit not as dramatically as ISC-specific silencing. In addition, we show that EB *spz* impinges on midgut mitosis upon infection. We have done these experiments and incorporated the results in new Fig. 1F and Fig. 2C. [Lines 172-176 and 226-229 respectively].

3- Is expression of the ligand Spz regulated by damage? Which cells produce this ligand?

Functional experiments have been conducted using RNAi knockdown in different cell types. However, the use of antibodies or reporter lines would be important to localize the source of the ligand.

Response: The source of the ligand is an important question. Using 2 independent RNAi lines, we have systematically silenced *spz* in different midgut cell populations, as well as tissues outside the gut. We found that there are multiple sources of Spz relevant here located in the midgut (progenitors, ISCs, EBs, ECs) and outside the gut (trachea and fat body). The new data are presented in new Fig. 2 [Lines 215 to 234].

4- How is Toll signalling activated in response to damage? In other words, what lies upstream of Spz/Toll activation in regeneration?

Response: Again, an interesting question but for a whole other project. One pathway regulating Toll in the gut, is Notch (see item B and G in the figure below). In *esg^{ts}; Notch-RNAi* flies there is overactivation of ISC mitosis and this needs an intact Toll/NF- κ B pathway, showing that normally, Notch suppresses Toll and ISC mitosis. However, two of our co-authors (Jun Zhou and Guofan Peng) have a paper submitted about this and it would be difficult for us to present these results in the present manuscript. See figure below, which I would ask you to keep confidential.

NOTE: We have removed unpublished data that had been provided for the referees in confidence.

5- Is there a role for Spz/Toll signalling in intestinal homeostasis? This has been tested using *esgts>toll* overexpression tools. However, clonal analysis over time, using gene loss of function should be done to address this question. Using *esg* flip out or MARCM system and staining with differentiation markers would be important.

Response: We have used several RNAi lines targeting multiple components (*Toll*, *dorsal*, *Dif*, *spz*, *PGRP-SA*) of the Toll pathway to show its necessity for ISC mitosis in homeostasis and regeneration upon damage (Fig. 1, Fig. 2, Fig. S1A-D). Our results clearly show that in homeostatic conditions the Toll pathway is necessary specifically in ISCs for mitosis and ISC number maintenance (Fig. 1B, E sucrose, Fig. S1A-C). We also show that activation of the Toll pathway, through *Toll^{10B}* or *Dif* overexpression, is sufficient to induce mitosis in the absence of damage (Fig. S1A-C). We do not believe MARCM clonal analysis is needed to prove that Toll/Spz/Dif/Dorsal is involved in intestinal ISC mitosis during homeostatic conditions. We do not believe MARCM clonal analysis is needed to prove that Toll/Spz/Dif/Dorsal is involved in intestinal ISC mitosis during homeostatic conditions.

6- The same applies to the data in Figure S10. Clonal analysis needs to be used in such setting.

Response: This is a control experiment to show that the specific type of JNK (the dominant negative form of *bsk*) does not influence ISC mitosis. It does not warrant MARCM clones. We recognise however, that we should re-phrase and instead of referring to the whole of the JNK pathway restrict the comment to JNK itself (i.e. *Bsk*). This has been done.

7- Related to the above point, the effect of Toll overexpression in cell differentiation should be clarified. In Figure S5 the authors show that overexpression of Toll signalling under the control of *esg-gal4* leads to increase in EE cells. On the other hand, they show that overexpression of Toll under *Su(H)-gal4* (EB specific *gal4*) leads to increase in EB cells (Figure 3B). This is not explored further. Is there a relationship between the EE and EB phenotype presented? Does gene loss of function in the same cell types produced the opposite result (in control or infected midguts).?

Response: We show that *Toll^{10B}* overexpression in progenitors leads to increased ISC mitosis (mitosis figures co-stain with *DI-lacZ*), as well as increased numbers of total gut cells including *esg+* progenitors (EBs and pre-EEs) and large mature ECs and EEs. However, the increase in EB cells we do explore further showing that it is a combination of more ISC mitosis and blocking of EB differentiation. The latter because there is an increase in EB cells (without EB mitosis) but with concomitantly increased ISC mitosis. This could only be explained if EBs were stopped from differentiating. [Lines 259-273].

8- Figure S2B: The expression pattern of *Dif* does not appear to overlap with *esg>RFP* cells but rather be complementary to it.

Response: We replaced this-perhaps confusing-figure with one which shows the expression in progenitor cells of several Toll pathway components as outlined in the figure below. See Fig. S2E.

NOTE: We have removed unpublished data that had been provided for the referees in confidence.

Fig. S2E. Toll/Spz/Cactus/NF- κ B/ expression in intestinal progenitors. Representative images of the midgut of flies with indicated genotypes expressing Spz, Toll, Cact-GFP, Dif-GFP or Dorsal-GFP (green) at 29°C for 10 days. Nuclei (blue), *esg>mcherry* (red). The white arrow shows GFP-positive cells.

9- What are the biological implications of reducing Toll signalling in the intestine. Analysis of organismal health and/or lifespan in homeostatic or infective settings would be important.

Response: We conducted lifespan assays with Toll-RNAi and Toll-10B flies in homeostatic conditions and following infection. [Lines 202-214 and new Figure S2].

Minor comments

1- There are several graphs with missing data (Figures 1B, S1A, D).

Response: Apologies for this. We have rectified the figures.

Reviewer 2

In their paper, "Toll signalling controls intestinal regeneration in *Drosophila*" the authors provide fairly convincing evidence that activation of the Toll pathway is necessary and sufficient for ISC mitosis and that JNK and AKT signalling are downstream of Toll activation. As the authors point out, the role of the Toll pathway in the fly midgut has been understudied. The author's work is an important start to establishing a role for the pathway. However, as it stands, I have some major concerns.

1) The overreliance on RNAi knockdown using *esg-Gal4* makes it difficult to determine what the phenotypes are exactly. A lot of the phenotypes seen could result from, as the authors put it, "massive" increases in cells and subsequent induction of cell death and injury. Rare, marked clones, as suggested below, would help to obtain more precise conclusions about the role of Toll signalling. **Response:** We disagree with this assertion. We complement RNAi assays with gain of function of Toll and epistatic analyses to understand what the phenotypes are. Crucially, let's not move away from what we want to say in this work: that Toll/NF- κ B is necessary and sufficient to promote ISC mitosis through Akt and JNKK. It is a relatively simple point that we think is made well with the assays used. Moreover, since the first review of this paper there have been two papers in PLoS Biol making a similar point in the wing and eye disc respectively in the context of *ras*^{V12} tumours (without the downstream transcriptional links to JNKK and Akt) that do not use MARCM clones (although one of them uses tumour clones but this is not applicable to our case).

2) It is unclear to me, after reading this paper, what the purpose of Toll signalling is. The Discussion section is mostly a rehash of the results. It would help to have some insight into why there is a need for this pathway to drive cell division when there are many others that can do this. **Response:** We have spent some considerable space in the discussion explaining what we think Toll is doing by trying to synthesise the results of activating ISC mitosis and increasing microbiota density (lines 443-467). We say that Toll is a regulator of the intestinal ecosystem, and we would urge the reviewer to re-read this part of the discussion.

SUGGESTIONS TO AUTHORS

Introduction:

- 1) Line 46: "was driving" should be "drives" [now line 47].
- 2) The authors should strive to cite more primary papers in the manuscript. While it is okay to include reviews, it makes it difficult for readers to determine which primary papers established the points the authors are trying to make [**tried to do this**].
- 3) Line 93: Micchelli and N. Perrimon 2006 does NOT use Delta to identify ISCs [**correct; it was Ohlstein & Spradling 2006, Nature 439**].
- 4) Line 103: Ohlstein and Spradling 2006; Ohlstein and Spradling 2007 do not mention *esg*. Better to cite: Micchelli and N. Perrimon 2006 [**we rectified this**].
- 5) Line 110-111. "as the posterior midgut is more immunologically active." My impression is that bacteria are found in the anterior. Could the authors provide evidence for the posterior midgut being more active? Also, in the paper the authors characterize the whole gut.

Response: We made these changes and tried to use more primary literature as the reviewer requests. As for the immunologically active domain of the gut this is from Nicolas Buchon's work in the Lemaitre lab (especially the Cell Host & Microbe papers in 2009 and 2012) as well as the papers on Duox by Won-Jae Lee's lab. We are looking at whole guts and we therefore deleted this phrase.

Results section:

- 6) The authors should list references for the genetic lines they use in the results section.

Response: This has been done. **Lines 138-144.**

7) Myo1A becomes expressed in ISC with age.

Response: We are using young (3-10 day old) flies.

8) Figure 1B. The data (bars) are missing from the graph. Same with Figure S1 and S1D.

Response: We apologies for this, this was rectified.

9) I'm surprised there are any mitosis in flies reared on sucrose, which lacks protein.

Response: Sucrose elevates circulating glucose, activating IIS (via Dilp2/3), which promotes ISC division see O'Brien et al., 2011 *Nature*, 476(7358), 63-68.

10) The Dif RNAi result does not look that different from the control in figure 1. Did the authors validate the line?

Response: Differences are statistically significant; all RNAi lines have been validated.

11) It would help the author's case if the authors made marked clones of Toll pathway mutants, both LOF and GOF. This would allow them to use validated alleles (nulls and hypomorphs) and to examine clone growth and composition. This would give definitive data regarding division rates and cell fate choices. I would suggest they do this with UAS-spz as well.

Response: This would help if our goal was to validate a Toll allelic series or focus on the source of Spz. As it is, the message of our work is to simply say that Toll is necessary and sufficient for ISC mitosis through Akt and JNKK as the title of the manuscript suggests. Please also see responses to points 5-6 for reviewer 1.

12) "Figure S1. Toll is not required in mature intestinal cells for midgut mitosis" This it is not an appropriate conclusion. First of all, the RNAi may not come on until after the EC has been determined. Second of all, the authors provide no firm evidence that the protein activity of these genes are affected by RNAi expression. They should just temper their statement.

Response: We have done this.

13) The authors should be cautious with conclusions using ISCs. The RNAi expressed in the ISC might be inherited by the EB and cause its effects there.

Response: As with item 12, statement has been tempered.

14) Figure S2B. What is the cell with green Dif staining but no mCherry (Left arrow) staining? How many cells become positive after infection? A low mag view would be helpful. Do the authors have any insight into which progenitor is positive for Dif? The ISC or the EB?

Response: See response to reviewer 1 and new Figure S1E showing Toll/Spz/NF- κ B/Cactus expression in progenitors without being able to say if it is ISCs or EBs or both.

15) Line 173. "into the its native locus (Fig 2C)". I think they mean FigureS2C.

Response: The reviewer is right. Apologies for the mix up, this was rectified (now Fig. S2E)

16) Figure S3 is confusing. If ISCs is a marker of ISCS then it should be roughly evenly distributed throughout the intestine in w1118. The data makes me think that it might mark a subset of ISCs, in some state of the cell cycle. Perhaps ones that are getting ready to divide. This would make sense, given that the Toll10B and UAS-DIF flies have more PH3 and more GFP cells. Otherwise, I would have to conclude spz, Dif, and PGRP RNAi flies have almost no stem cells. If so, the authors need to do an appropriate characterization of this phenotype. Are ISCs dying or differentiating? In any event, the authors need to clarify this.

Response: This is an interesting question. Even if in the absence of Toll signalling there is practically no ISC mitosis, the ISCs are there and able to divide. We know this from our previous work [Bahuguna et al., 2022, *PLoS Genet* 18(1), e1009992]. There, we found that in *pgrp-sa* or *dif*, loss of function whole-body mutants, ISC mitosis and ISC numbers were extremely low. However, treatment of flies with rapamycin (or RNAi of *mTOR*) re-activated ISC mitosis in the absence of Toll. We interpreted this as Toll being able to sustain a metabolic state that enables ISC mitosis in the gut but also nourish the microbiota (as when *pgrp-sa* or *dif* are absent cultivable bacterial density drops to almost zero as well-something that contributes to the mitotic silence of ISCs, Bahuguna et al., 2022, *PLoS Genet* as above).

17) I worry that constitutive activation of the Toll pathway could create artifacts. The authors should look at PH3 after heat induction of UAS-spz.

Response: Not sure why this would help distinguish between artifacts and real effects. Also note that we show that *UAS-Dif* increases mitosis and tends to increase ISC numbers (Fig. S1A-C). In fact, *UAS-spz* overexpression will have non-autonomous effects, since it is a secreted ligand. As a reminder, all we want to show here is that Toll signalling is both necessary and sufficient to activate ISC mitosis.

18) The word "massive" is vague.

Response: We have changed the phrasing from "massive" to "significant".

19) It's unclear to me what the *esg*>GFP cells are in Toll10B. *esg* can mark enterocytes with stress, damage, and aging. Staining with Pdm1 could help with this. Also, as mentioned above, marked clones expressing spz or Toll10B would make it easier to assess the nature of the cells.

Response: These are not ECs. Firstly, because *esg*>Toll10B blocks EB to EC transition (with subsequent reduction of EC numbers over time) see Figs 3 and 4 and lines 259-284. Secondly, because none of the GFP+ cells in the *esg*>Toll10B condition are large polyploid cells.

20) Line 219: (Ohlstein and Spradling, 2007) did not use *Dl-lacZ*. I would also caution using this reporter as it has been shown that *Dl* can be expressed in non-ISCs with injury. Also, work from the Jiang lab suggests EBs can undergo mitosis.

Response: We are using the *Dl-lacZ* in homeostatic conditions and therefore no injury is at play here. In our hands when Toll is expressed in EBs there is no mitosis (Fig. 4C).

21) "The legend of Figure 3D and the text states "Activation of Toll in EBs did increase Su(H)+ cells." Where is the quantification for this conclusion?

Response: No quantification in this case, but representative images in Fig. 4D.

22) Figure S6A (Toll10B, *Dif* RNAi) is not consistent with the quantification of the data in the C and D. Looks like the decrease in GFP cells is partial compared to w1118.

Response: Fig S6A is a snapshot of a trend that data in D is consistent with. They suggest a (non-significant but still indicative) trend that shows more GFP+/cells ratio in the Toll10B, *Dif* RNAi. However, since this is not significant in the quantification it is not appropriate to call it a "partial" rescue.

23) Figure S7A. A region appears negative for GFP. What region do the authors think this is?

Response: This panel has been taken out of the current version to consolidate Fig. S7. As to not avoid the response we think it is R5 (day 10) and R3 (day 22). However, these regions are not negative, but GFP+ cells are markedly reduced.

24) What was the rationale to look at JNK signalling by RNA-seq?

Response: By definition, RNA-seq looks at global changes. Within these we picked up JNKK as one major transcriptional target. We then hypothesised that NF- κ B-dependent transcriptional activation of JNKK would lead to JNKK/JNK-dependent transcription hence tested *puc-lacZ* a marker for JNK transcriptional activity.

25) Line 324-325: "These data indicate that JNK signalling is not necessary for the maintenance of epithelial turnover in homeostatic conditions." It's more accurate to say it is not necessary for cell division in homeostatic conditions." Has this not been addressed by the Jasper lab?

Response: We changed the phrasing, to "these data indicate that expression of the negative form of JNK in intestinal progenitors does not influence epithelial turnover in homeostatic conditions." [Lines 365-367].

26) What do the authors mean by "epithelial integrity" in line 328?

Response: Epithelial integrity: "The balance between ISC mitosis, EB production and ultimately the generation of ECs". We have re-phrased.

Reviewer 3:

Udayakumar et al. undertook the advantage of *Drosophila* genetic tractability to show in detail and confirm consistently that the Spz/Toll/NF- κ B/Dif signalling axis is required to promote Intestinal Stem Cells (ISC) mitosis. In addition, they connect Toll activation with the increase of gut bacterial density. They further show Toll activity was mediated by JNK and Akt/mTOR signalling, giving more insights into ISC regulation. The Toll control of intestinal homeostasis is an important discovery of general interest. Overall, the work is well-performed and includes a variety of experimental approaches. However, there are certain concerns that remain to be addressed.

SUGGESTIONS TO AUTHORS

1. This work nicely demonstrates the role of Toll pathway in the control of stem cell proliferation under homeostasis and infection. However, an important question that remains to be addressed is: what is the relevance of the identified mechanism? How will knockdown of Toll and subsequent epithelial regeneration affect fly survival under homeostatic conditions (and aging), and will they survive infection differently in comparison to controls? In another words, will impair epithelial renewal due to Toll pathway suppression make flies more susceptible to infections?

Response: We have done these experiments and present the data in new Fig. S2.

2. It would be useful to include as supplementary table the full list of DEGs and not only the top 20.

Response: We now provide this information (Tables S2-S10).

3. Regarding the presentation of data, I would ask the authors to include missing box plot data from Figure 1B, Figure S1A, and Figure S1D, and for those graphs, do the statistical comparison of P.a.-treated flies vs sucrose-treated flies. Otherwise, the separation of data to analyze the comparison of RNAi to control flies within the same treatment, like in Figure 1 and Figure S1, is redundant.

Response: Apologies for this omission, which was probably due to the word-to-pdf conversion but was not immediately obvious when we inspected the pre-submitted manuscript. Also note that we have removed the graphs showing sucrose and P.a.-infected flies separately to avoid redundancy.

4. Regarding the statistical analysis and interpretation of data, here are my specific observations and concerns:

- Currently, if understood correctly, Figure 1B should have the same data and analysis as Figure 1C and 1D - so double check whether spz RNAi suc is * or ** significant.

Response: See new Figure 2 with extended *spz* data to pinpoint the source of the ligand in both sucrose and P.a.-infected conditions. [Lines 215-234].

- Line 322 In 'and the proportion of GFP+ cells (Fig. S10B) were statistically indistinguishable from that of the control. On comparing the number of pH3+ cells in *esgts-GAL4* controls and *esgts-GAL4; UAS-bskDN* guts, there was a small reduction in pH3+ cell numbers ($p < 0.05$)' - Please, exclude or clarify whether Fig. S10B was statistically indistinguishable from that of the control, since the plot shows significant differences.

Response: It is marginally significant at the level of pH3+ cells but not the level of total cells and therefore we consider it not physiologically relevant.

- Figure 5B ESG-ts is a control, so it should go first, and asterisk lines should be clarified (why is it only one long line?).

Response: It was drawn as such for convenience as the *esgts-GAL4; UASToll10B* is equally different from all three other genotypes (hence the long line indicating the same difference from each). Thus, we did not change the graph.

- The authors are advised to make their graphs consistent, for example Fig. 2B is different vs 2C-F. Why not to use uniform graph format?

Response: Old Fig. 2 is now Fig. 3. Old Fig. 2B was GraphPad whereas 2C-F is the output graph from the imagining plug-in program we have created

(https://github.com/jefflee1103/Imageanalysis/tree/main/people/Ash_U) see lines 589-598. Same for the graphs in Fig. 3. We did not change these outputs.

- Figure S2A should be improved. First, as I understand each genotype has uninfected and infected conditions - those should be clearly marked on the graph. Second, the aim of this graph is to compare Toll RNAi treatments with w1118 after Ecc15 infection. Hence, these comparisons should be shown on the graph. Third, include the driver label (ISCts) on the plot.

5. To obtain clarity and easier reading for wider audience, I recommend including the explanation of used fly lines when they are first mentioned in the text. Examples:

- EE-GAL4 is mentioned in the results (Line 150) but was not included when mentioning all GAL4 lines (Line 139). **[Now mentioned in line 141].**

- esgts-GAL4/UAS-Toll10b; UAS-bskDN is first mentioned and used in the experiment in Line 259 but explained only later in text in Line 307 (...also overexpressed a dominant negative form of JNK (UAS-basketDN, or UAS-bskDN...)). **[This has been rectified, see lines 298-299].**

-Elaborate the usage/mentioning of tak1 (Line 298). **[VDRC KK line 101357]. Now mentioned in line 493.**

Response: We have implemented these recommendations.

In addition, for clarity in the methods part, either put PA14 protocol before Ecc15, or already mention that female mated flies were used for all experiments.

6. Again, to obtain clarity and easier reading for the wider audience, I would recommend including a schematic of gut cells (something like: https://www.researchgate.net/figure/Proliferation-and-differentiation-of-intestinal-stem-cell-ISC-in-Drosophila-The_fig1_376715092) and inclusion of GAL4 lines that you used to test them in this paper.

Response: We have implemented some of these recommendations **[Line 497-506]**. To consolidate the manuscript, we removed Ecc15 infection and replaced it with DSS stress.

Specific recommendations:

Figure 6B include the driver label on the plot and figure description

Line 451 include the Rifampicin concentration

Line 573 write the volume of PBS that was used

Line 66 D-V add 'dorsoventral' before abbreviation

Line 188 not so relevant here: Drosophila ISC numbers increase with age leading to dysplasia in older guts - more connected later to Line 257

Response: We have implement these recommendations.

Second decision letter

MS ID#: dev.204794R1

MS Title: Toll signalling controls intestinal regeneration in Drosophila

Authors: Aiswarya Udayakumar; Filippou Stavropoulos; Theodosia Hadjipanteli; Guofan Peng; Shivohum Bahuguna; Caitlin MacClay; Jeffrey Y. Lee; Qi Xiao; Yuxian Xia; Michael Boutros; Jun Zhou; Yiorgos Apidianakis; Chrysoula Pitsouli; Petros Ligoxygakis

Article Type: Research Article

Dear Dr Ligoxygakis,

I have now received all the referees reports on the above manuscript, and have reached a decision. The referees' comments are appended below, or you can access them online: please go to .

The overall evaluation is positive and we would like to publish a revised manuscript in Development, provided that the referees' comments can be satisfactorily addressed. Please attend to all of the reviewers' comments in your revised manuscript and detail them in your point-by-point response.

Reviewer 1

The authors have addressed my main comments and the paper is suitable for publication.

Reviewer 2

SUMMARY OF THE ADVANCE MADE IN THIS PAPER AND ITS POTENTIAL SIGNIFICANCE TO THE FIELD

The authors have responded to my concerns. I support its publication in Development.

Reviewer 3

SUMMARY OF THE ADVANCE MADE IN THIS PAPER AND ITS POTENTIAL SIGNIFICANCE TO THE FIELD

Udayakumar et al. undertook the advantage of *Drosophila* genetic tractability to show in detail and confirm consistently that the Spz/Toll/NF- κ B/Dif signalling axis is required to promote Intestinal Stem Cells (ISC) mitosis. In addition, they connect Toll activation with the increase of gut bacterial density. They further show Toll activity was mediated by JNK and Akt/mTOR signalling, giving more insights into ISC regulation. The Toll control of intestinal homeostasis is an important discovery of general interest. Overall, the work is well-performed and includes a variety of experimental approaches.

SUGGESTIONS TO AUTHORS

The authors addressed my comments in the revised manuscript. However, the added survival experiments require clarifications. First, it should be described in Methods how they were performed. Second, the rationale of using *S. aureus* for intestinal infections is unclear. *S. aureus* is an unusual pathogen for *Drosophila* intestinal infections. *P. aeruginosa* would be more appropriate for such infections. First, it is a well-established and commonly-used pathogen for *Drosophila* intestinal infections. Second, for consistency with all other experiments, as the authors used *P. aeruginosa* for them. Finally, the fact that it is not known whether *S. aureus* induces Toll-dependent epithelial renewal, questions the suitability of this pathogen for addressing the relevance of Toll-dependent epithelial renewal during infection. Hence, survivals with *P. aeruginosa* would make the manuscript more coherent. Also, making survival graphs bigger would improve their readability.

Figure 5 lacks panel labels

Second revisionAuthor response to reviewers' comments-----
Reviewer 3:

Udayakumar et al. undertook the advantage of *Drosophila* genetic tractability to show in detail and confirm consistently that the Spz/Toll/NF- κ B/Dif signalling axis is required to promote Intestinal Stem Cells (ISC) mitosis. In addition, they connect Toll activation with the increase of gut bacterial density. They further show Toll activity was mediated by JNK and Akt/mTOR signalling, giving more insights into ISC regulation. The Toll control of intestinal homeostasis is an important discovery of general interest. Overall, the work is well-performed and includes a variety of experimental approaches.

SUGGESTIONS TO AUTHORS

The rationale of using *S. aureus* for intestinal infections is unclear. *S. aureus* is an unusual pathogen for *Drosophila* intestinal infections. *P. aeruginosa* would be more appropriate for such infections. First, it is a well-established and commonly used pathogen for *Drosophila* intestinal infections. Second, for consistency with all other experiments, as the authors used *P. aeruginosa* for them.

Response: We strongly disagree with the premise of this argument. Using just one pathogen for all experiments is exactly the kind of **consistency we do not want**. Time and again in the fly immunity field, people have used one pathogen throughout a study and came to a sweeping conclusion that has been subsequently disputed or dismantled. One example is Leulier *et al.* Nat Immunol 2003 where all experiments were done with *Micrococcus luteus* as a representative of Gram-positive bacteria. They made sweeping statement about how Gram+ bacteria are recognised by Toll and Gram-negative by IMD. Well, it turns out that *M. luteus* is not representative of Gram-positive bacteria and has in fact a rather obscure peptidoglycan structure that may obscure results. See Vaz *et al.*, Cell Reports 2019 for the arguments of Toll/IMD activation when a truly representative of 95% of Gram-positive bacteria is used namely...*S. aureus*. Another such case is Leclerc *et al.*, EMBO Reports 2006, where just by using again 1-2 pathogens and just one *Drosophila* Prophenoloxidase (PO) mutant they made a sweeping statement about PO not needed to fight off infection. This has now been disputed by several papers in *Drosophila* as well as other insects. We wanted therefore to see if another pathogen, one that is representative of Gram-positive bacteria will be able to kill Toll mutants in systemic (injection) and enteric (feeding) infection.

Finally, the fact that it is not known whether *S. aureus* induces Toll-dependent epithelial renewal, questions the suitability of this pathogen for addressing the relevance of Toll-dependent epithelial renewal during infection. Hence, survival with *P. aeruginosa* would make the manuscript more coherent. Also, making survival graphs bigger would improve their readability.

Response: We have now provided data that show that there is Toll-dependent expansion of progenitor cells following *S. aureus* enteric infection. See new Fig S2C and S2D. We also changed the graph presentation to make them more readable.

Figure 5 lacks panel labels

Response: This has been rectified.

Third decision letter

MS ID#: dev.204794R2

MS Title: Toll signalling controls intestinal regeneration in *Drosophila*

Authors: Aiswarya Udayakumar; Filippou Stavropoulos; Theodosia Hadjipanteli; Guofan Peng; Shivohum Bahuguna; Caitlin MacClay; Jeffrey Y. Lee; Qi Xiao; Yuxian Xia; Michael Boutros; Jun Zhou; Yiorgos Apidianakis; Chrysoula Pitsouli; Petros Ligoxygakis
Article Type: Research Article

Dear Dr Ligoxygakis,

I am happy to tell you that your manuscript has been accepted for publication in *Development*, pending our standard publication integrity checks.